# Lonafarnib improves cardiovascular function and survival in a mouse model of Hutchinson-Gilford progeria syndrome

Sae-Il Murtada[1], Nicole Mikush[2], Mo Wang[3], Pengwei Ren[3], Yuki Kawamura[1], Abhay B Ramachandra[1], David S Li[1], Demetrios T Braddock[4], George Tellides[3,5], Leslie B Gordon[6], Jay D Humphrey[1,5]*

[1]Department of Biomedical Engineering, Yale University, New Haven, United States; [2]Translational Research Imaging Center, Yale University, New Haven, United States; [3]Department of Surgery, Yale University, New Haven, United States; [4]Department of Pathology, Yale University, New Haven, United States; [5]Vascular Biology and Therapeutics Program, Yale University, New Haven, United States; [6]Department of Pediatrics, Hasbro Children's Hospital and Warren Albert Medical School, Brown University, Providence, United States

**Abstract** Clinical trials have demonstrated that lonafarnib, a farnesyltransferase inhibitor, extends the lifespan in patients afflicted by Hutchinson-Gilford progeria syndrome, a devastating condition that accelerates many characteristics of aging and results in premature death due to cardiovascular sequelae. The US Food and Drug Administration approved Zokinvy (lonafarnib) in November 2020 for treating these patients, yet a detailed examination of drug-associated effects on cardiovascular structure, properties, and function has remained wanting. In this paper, we report encouraging outcomes of daily post-weaning treatment with lonafarnib on the composition and biomechanical phenotype of elastic and muscular arteries as well as associated cardiac function in a well-accepted mouse model of progeria that exhibits severe perimorbid cardiovascular disease. Lonafarnib resulted in 100% survival of the treated progeria mice to the study end-point (time of 50% survival of untreated mice), with associated improvements in arterial structure and function working together to significantly reduce pulse wave velocity and improve left ventricular diastolic function. By contrast, neither treatment with the mTOR inhibitor rapamycin alone nor dual treatment with lonafarnib plus rapamycin improved outcomes over that achieved with lonafarnib monotherapy.

*For correspondence:
jay.humphrey@yale.edu

## Editor's evaluation

The authors performed an important study to demonstrate efficacy of lonafarnib, the only approved drug for treating progeria in patients, using a well-established mouse model of the disease. The authors provided convincing evidence that lonafarnib extends lifespan in these mice that is associated with reduced arterial thickening and improved arterial function in an experimental model progeria. This paper is of major interest to scientists and clinicians interested in cardiovascular physiopathology, aging, and progeria in particular.

## Introduction

Hutchinson-Gilford progeria syndrome (HGPS) is an ultra-rare condition that results from a de novo autosomal dominant mutation to the gene (*LMNA*) that codes the nuclear envelope scaffolding protein lamin A (*De Sandre-Giovannoli et al., 2003*; *Eriksson et al., 2003*), which results in intranuclear

accumulation of an aberrant form of the protein called progerin. This accumulation results from a variant that lacks a post-translational proteolytic processing step, leaving the progerin protein permanently modified by a farnesyl group. Clinical trials show that farnesyltransferase inhibitors extend healthspan and lifespan in children afflicted with HGPS (*Gordon et al., 2012*; *Gordon et al., 2014*; *Gordon et al., 2018*), and the US Food and Drug Administration approved on November 20, 2020 the use of the Zokinvy (lonafarnib) to slow the accumulation of progerin with the hope of extending lifespan (*Dhillon, 2021*).

Although HGPS is characterized by wide-spread tissue and organ abnormalities, premature death appears to result from cardiovascular complications, including atherosclerosis-related heart failure, with progressive left ventricular (LV) diastolic dysfunction as the most prevalent condition observed in a recent clinical study (*Prakash et al., 2018*). It is well known in the normal aging population that increased aortic stiffness, and attendant increases in central pulse wave velocity (PWV), contributes significantly to both atherosclerosis and diastolic dysfunction/heart failure with preserved ejection fraction (*Abhayaratna et al., 2006*; *van Popele et al., 2006*; *Desai et al., 2009*; *Townsend et al., 2015*). Importantly, marked increases in ankle-brachial and carotid-femoral PWV have been reported in children with HGPS (*Gerhard-Herman et al., 2012*) and early results suggested that lonafarnib can lessen this increase (*Gordon et al., 2012*).

Notwithstanding the promise of lonafarnib in extending the lifespan of HGPS patients, the largest and most recent clinical study evaluated mortality, comparing treated trial patients with a contemporaneous untreated control group as its primary outcome (*Gordon et al., 2018*). That is, the effects of lonafarnib on central artery stiffness and associated cardiovascular function have not been examined in detail. In this paper, we quantify and compare diverse functional cardiovascular metrics, including elastic and muscular artery stiffness and vasoactivity as well as cardiac performance, in untreated and lonafarnib-treated $Lmna^{G609G/G609G}$ progeria mice (denoted as GG in figures). The original report of this mouse model indicated 50% survival at 242 days of age in the heterozygous ($Lmna^{G609G/+}$) animals but only 103 days in the homozygous ($Lmna^{G6096/G609G}$) animals (*Osorio et al., 2011*). All final assessments of structure and function herein for littermate wild-type ($Lmna^{+/+}$, or WT) and progeria ($Lmna^{G609G/G609G}$) mice are at 168 or 169 days of age, with increased survival of untreated progeria mice achieved via an improved feeding and care protocol that allowed the cardiovascular phenotype to worsen further, similar to that observed clinically in patients. Importantly, daily post-weaning treatment with lonafarnib improved cardiovascular function and yielded 100% survival to the study end-point of 168 days. Inhibitors of the mechanistic target of rapamycin (mTOR) have also shown some promise in vitro in treating progeria cells (*Cao et al., 2011*; *DuBose et al., 2018*), thus we also examined the potential benefits of late-term daily injections of rapamycin. Neither treatment with rapamycin, an inhibitor of mTOR, nor combination therapy of lonafarnib plus rapamycin improved outcomes over lonafarnib alone though these findings must be interpreted cautiously.

## Results

### The aortic phenotype worsens rapidly in the perimorbid period in progeria

Switching from normal chow placed on the floor of the cage at weaning to a soft gel-based hydrated chow placed on the floor at weaning (i.e. postnatal day P21) coupled with the introduction of a caretaker mouse in each cage extended the mean survival of untreated progeria mice by an additional ~12%, from ~150 (*Murtada et al., 2020*) to ~168 days of age, at which time cardiovascular function was evaluated. *Figure 1A* compares standard passive pressure-diameter responses (reflecting the underlying microstructural composition and organization) of the descending thoracic aorta (DTA) in wild-type (WT) littermate and untreated progeria (GG) mice at increasing ages. There was a progressive structural stiffening in progeria (i.e. a left-ward shift of the pressure-diameter response) from postnatal day P42 to P140, with a remarkable stiffening thereafter to P168. This late marked stiffening is associated with a grossly translucent, brittle appearance (*Figure 1B*), reflecting an extreme diffuse calcification.

Quantification of circumferential material stiffness (*Figure 1C*) and geometry (*Figure 1D and E*) enables calculation of the local aortic pulse wave velocity via the Moens-Korteweg relation, $PWV = \sqrt{Eh/2\rho a}$ , where $E$ is material stiffness, $h$ wall thickness, $a$ luminal radius, and $\rho$ mass density of the blood. Circumferential material stiffness was numerically lower in the progeria aorta at P42 relative

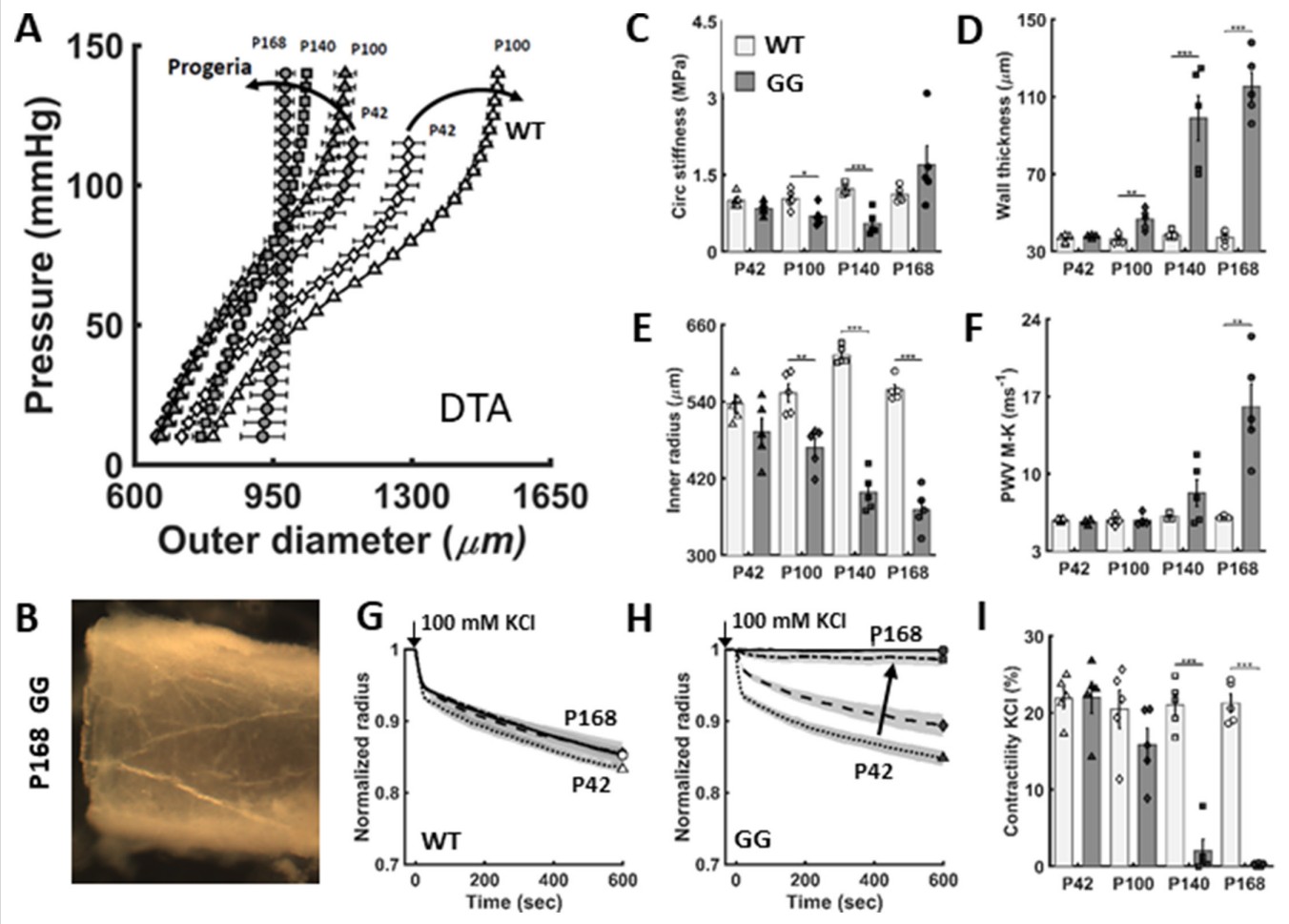

**Figure 1.** The biomechanical phenotype of the descending thoracic aorta (DTA) worsens progressively in untreated *Lmna*^G609G/G609G progeria (GG) mice relative to age-matched wild-type (WT) littermate mice. (**A**) Pressure-diameter responses revealed a progressive structural stiffening in progeria from postnatal day P42 to P100 to P140 to P168 (dark curved arrow). (**B**) The marked, late-stage aortic stiffening is associated with a translucent appearance and brittle texture. (**C–F**) Passive geometric and mechanical metrics calculated at a common distending pressure of 100 mmHg for the DTA in untreated progeria mice (darker bars) confirmed marked progressive worsening relative to wild-type (light bars). (**G–I**) Vasoactive capacity, revealed as diameter reduction in response to high potassium depolarization of the smooth muscle cell membrane (100 mM KCl), first as a function of time and extent of response and second as steady-state response at 10 min, both for different ages. Note the near complete loss of contractility by P140 and its complete loss by P168 days in progeria. Here, n=5 vessels per group, with *, **, and *** denoting statistical significance, between WT and GG at the age indicated, at p<0.05, p<0.01, and p<0.001, respectively. See also *Figure 1—figure supplement 1* for additional biomechanical metrics as well as *Source data 1 and 2* for all numerical values.

The online version of this article includes the following figure supplement(s) for figure 1:

**Figure supplement 1.** Additional biaxial metrics confirm the progressive worsening of the aortic phenotype.

to that in age-matched wild-type controls (0.84 MPa in progeria, 1.13 MPa in WT), but higher at P168 (1.69 MPa in progeria, 1.11 MPa in WT). This late increase, in combination with a marked increase in wall thickness (from 39 μm at P42 to 115 μm at P168 in progeria) and decrease in luminal radius (from 474 μm at P42 to 371 μm at P168 in progeria), significantly increased the calculated local PWV (*Figure 1F*) by more than threefold in progeria at 168 days of age (from 5.3 m/s at P42 to 16.2 m/s at P168). The vasoconstrictive capacity of this segment of the progeria aorta was also markedly abnormal (effectively absent) in the perimorbid period, with loss of contractile capacity seen by P140 relative to control (*Figure 1G–I*). Additional biomechanical metrics for the DTA, including biaxial, are contrasted in *Figure 1—figure supplement 1* for wild-type and progeria mice, noting that biaxial wall stretch and stress were markedly lower in progeria, and so too elastic energy storage capacity, especially in the perimorbid period. Because storage of elastic energy during systolic distension normally facilitates

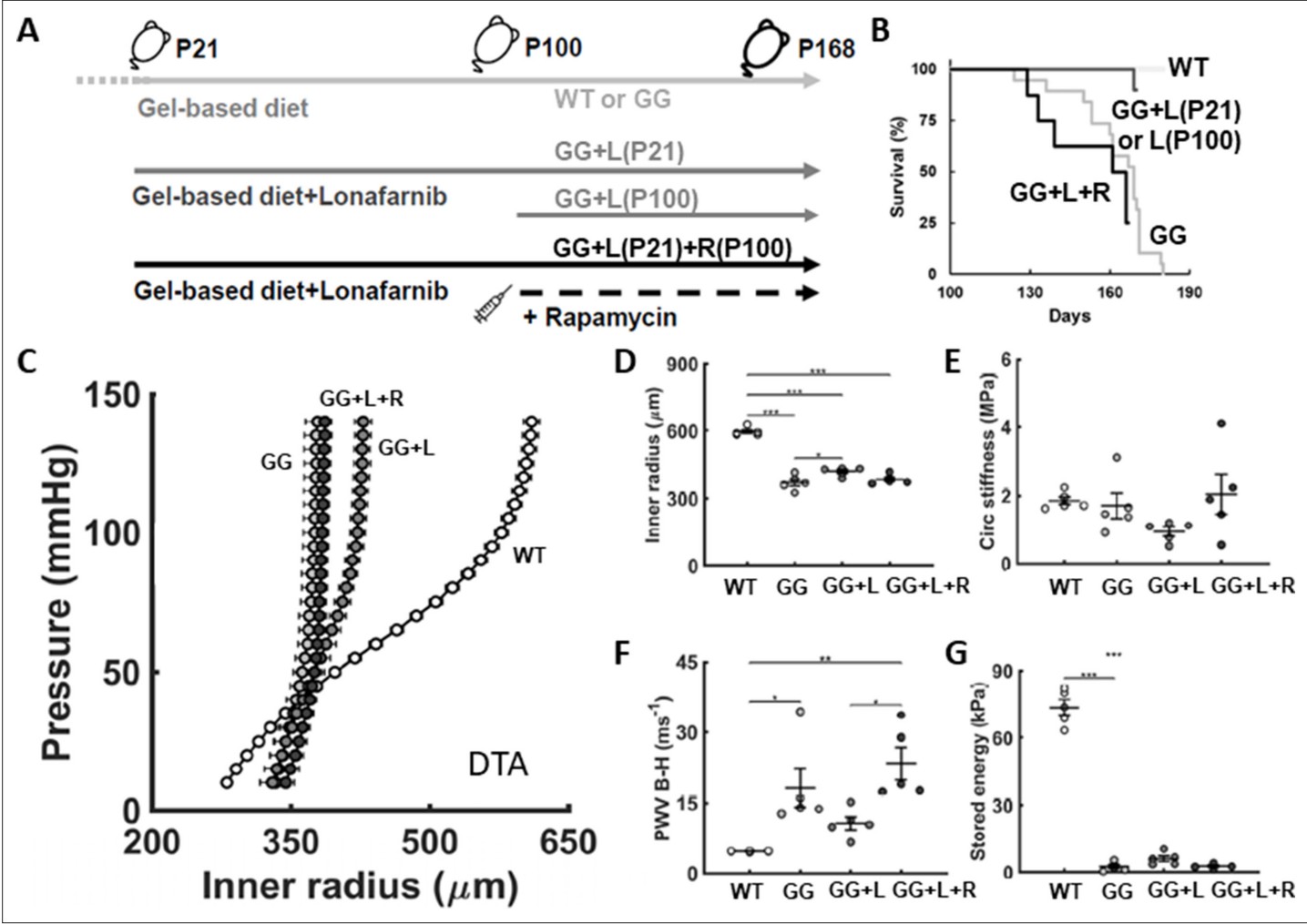

**Figure 2.** Lonafarnib improves survival and central artery stiffness in progeria mice (GG). (**A**) Study design, including untreated progeria mice and progeria mice that were given lonafarnib daily in the soft gel-based chow either from postnatal day P100 to P168, denoted L(P100), or from P21 to P168, denoted L(P21), and finally L(P21) plus rapamycin R(100) from P100 to P168. Soft gel-based chow, without or with the drug, was started at weaning (P21) in all mice. (**B**) All 10 (100%) of the lonafarnib-treated progeria mice, namely 4/4 of the GG + L(P100) and 6/6 of the GG + L(P21) mice, survived to the intended end-point, P168, while only 10 of the 19 untreated GG mice (~53%) survived to P168; note that the one GG + L death occurred at P169, the day after anesthesia, echocardiography, and recovery while three additional untreated GG mice died by P169. The five untreated wild-types (WT) littermate controls all survived to the study end-point, as expected, whereas lonafarnib plus rapamycin did not improve the survival of the progeria mice. Consistent with the improved survival for lonafarnib treatment from P21, the structural stiffness of the descending thoracic aorta (DTA) improved as revealed by both (**C**) standard pressure-diameter testing ex vivo and (**F**) local pulse wave velocity (PWV) calculated using the Bramwell-Hill (B-H) equation. (**D,E,G**) There was, however, no improvement in intrinsic metrics at physiological conditions. Here, n=4 (lonafarnib from P100) or 5 (all other groups) vessels per biomechanical testing group, with *, **, and *** denoting statistical significance at p<0.05, p<0.01, and p<0.001, respectively. See also *Figure 2—figure supplement 1* as well as *Source data 1–3* for all numerical values.

The online version of this article includes the following figure supplement(s) for figure 2:

**Figure supplement 1.** Addtional biaxial findings confirm positive effects of lonafarnib on select metrics.

the elastic recoil during diastole that augments antegrade and retrograde blood flow, this marked loss of energy storage reveals a dramatic reduction in the biomechanical function of the aorta in progeria.

## Lonafarnib improves survival

Only 53% (n=10/19) of the untreated progeria mice survived to the time of scheduled cardiac function assessment at postnatal day P168. By contrast, 100% (n=10/10) of the progeria mice treated daily with lonafarnib (450 mg per kg of gel-based chow), either from the time of weaning at P21 (n=6/6) or from P100 (n=4/4), survived to P168 (*Figure 2A and B*). Note, however, that 1 of the 6 mice treated

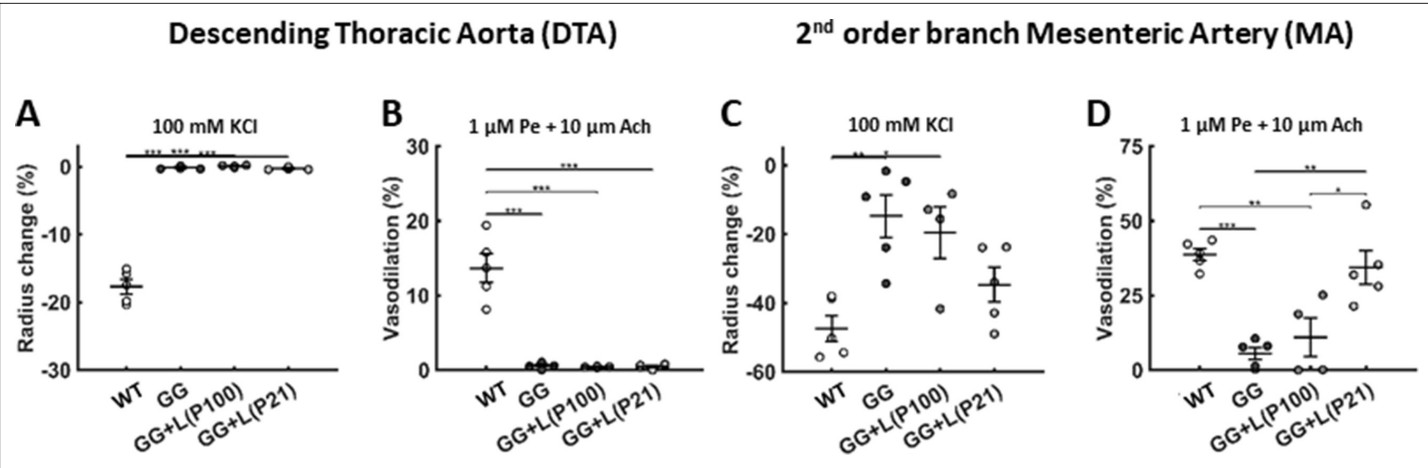

**Figure 3.** Lonafarnib treatment improves the vasoactive function of muscular but not elastic arteries. (**A–B**) The vasoconstrictive capacity of the aorta declined progressively in untreated progeria mice to near nonexistent levels by postnatal day P140, which persisted to P168, and lonafarnib treatment from P21 or P100 did not improve this function when evaluated at P168. (**C–D**) There was a similar progressive, though a less severe, decline in vasoconstrictive capacity of the second-order branch mesenteric artery in untreated progeria mice, notably at P168. This mesenteric artery function was improved by lonafarnib treatment from P100 and especially from P21 as revealed by improved vasoconstrictive and vasodilatory function at P168. Vasoconstriction was evaluated isobarically in response to 100 mM potassium chloride (KCl); vasodilatation was evaluated isobarically in response to 10 µM acetylcholine (Ach) following attempted pre-constriction with 1 µM phenylephrine (Pe). n=4 (lonafarnib from P100) or 5 (all others) vessels per group, with *, **, and *** denoting statistical significance at p<0.05, p<0.01, and p<0.001, respectively. See *Source data 2* for numerical values. Note that we did not assess mesenteric artery function in the lonafarnib plus rapamycin group given the poor survival outcome.

from weaning was found dead the day after anesthesia, echocardiography, and recovery whereas an additional 3 of the 10 untreated progeria mice that survived to day 168 were also found dead on day 169, thus resulting in 7/19 (36.8%) untreated progeria mice surviving to P169 versus 9/10 (90%) lonafarnib-treated progeria mice surviving to P169 despite the burden of isoflurane anesthesia and echocardiography on P168.

## Lonafarnib improves aortic composition and properties, but not vasoactive function

Treatment with lonafarnib from either P21 or P100 to P168 improved the structural stiffness of the aorta as revealed by both the pressure-inner radius data (i.e. modest right-ward shifts; *Figure 2C*) and (despite modest changes in inner radius and circumferential material stiffness, *Figure 2D and E*) the reduced values of pulse wave velocity (i.e. no longer statistically different from wild-type control values; *Figure 2F*), here computed independently using the Bramwell-Hill relation ($PWV = \sqrt{1/\rho D}$ where $\rho$ is again the mass density of the blood but $D$ is the clinical measure of distensibility, namely, $D = (d_{sys} - d_{dias}) / (P_{sys} - P_{dias}) d_{dias}$ where $d$ and $P$ are the inner diameter and distending pressure, respectively, and $sys$ and $dias$ denote systolic and diastolic conditions). This improved PWV at P168 emerged despite minimal improvement with lonafarnib treatment in circumferential wall stress (e.g. from 44 kPa in progeria to 59 kPa when progeria was treated from P21, both relative to a wild-type control value of 201 kPa) and elastic energy storage capability (from 2 to 6 kPa when treated from P21, relative to a wild-type control value of 60 kPa; *Figure 2G*), with no recovery of the vasoactive capacity of this segment of the aorta (*Figure 3A and B*).

Histological sections of the DTA from age-matched wild-type and untreated progeria mice at P168 revealed an expected marked decrease in medial smooth muscle cells (namely, a 79% reduction in medial cytoplasm area fraction in Movat staining, from 0.314 to 0.065), a decrease in medial collagen (a 52% reduction in medial area fraction, from 0.094 to 0.046), and dramatic increase in medial proteoglycans (a 3.9-fold increase, from an area fraction of 0.124 to 0.480) with progeria. Although the elastic laminae were largely intact, they appeared less undulated, likely due to the presence of excessive proteoglycans (*Figure 4A*), which increase intramural Gibbs-Donnan swelling pressures (*Murtada et al., 2020*). Focusing on the lonafarnib treatment from P21 to P168, histology revealed a slight improvement in smooth muscle area fraction (a 67% rather than 79% reduction), a near preservation

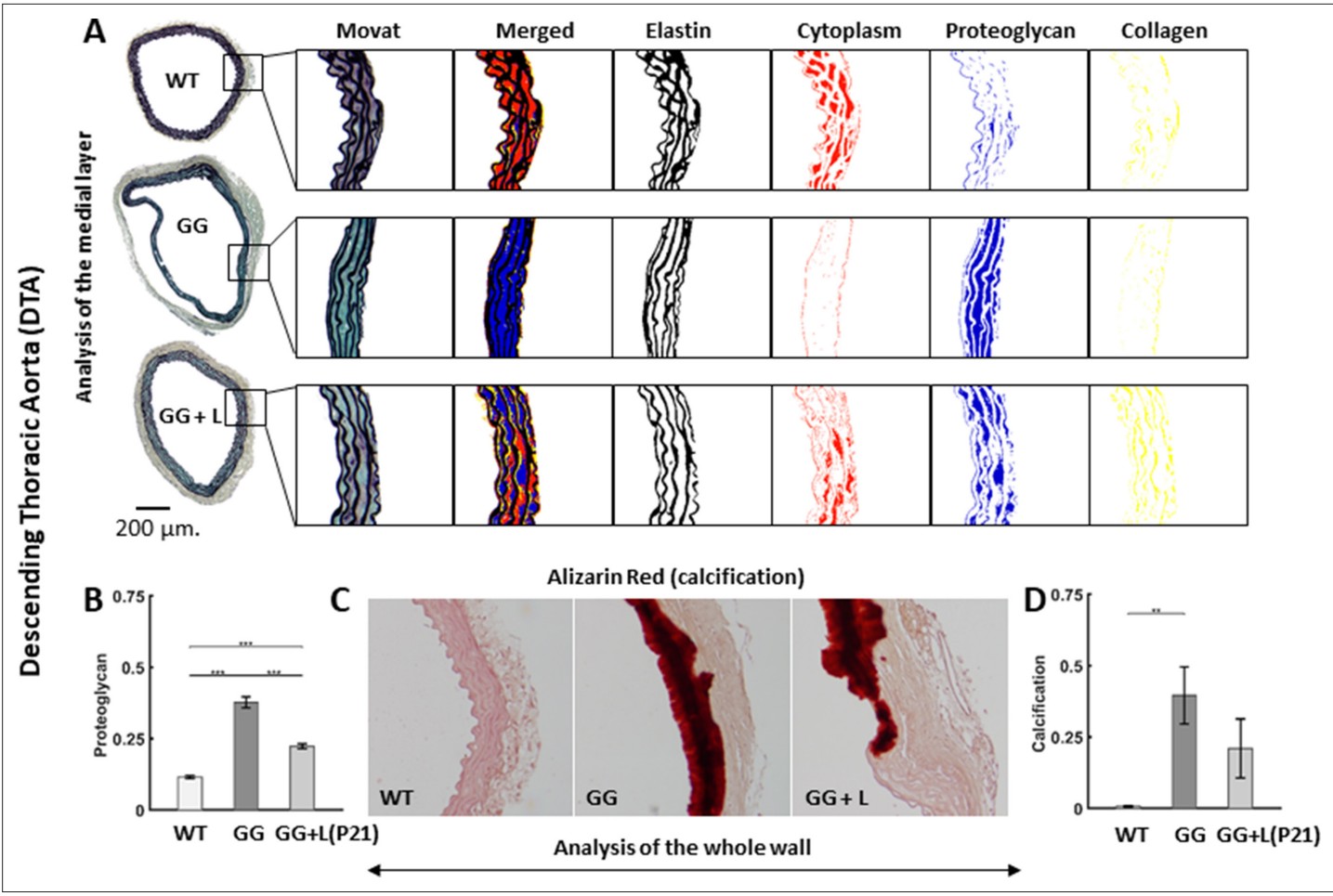

**Figure 4.** Lonafarnib treatment improves histological features in the descending thoracic aorta (DTA), with the drug given from P21 to P168.
(**A**) Representative Movat-stained cross-sections with computer-based automatic separation of elastic fibers (black), cell cytoplasm (red), proteoglycans/glycosaminoglycans (blue), and collagen (yellow) in the medial layer. Note the marked decrease in cytoplasm and increase in proteoglycans/glycosaminoglycan in untreated progeria (GG) relative to wild-type controls (WT), which is partially prevented by lonafarnib (GG + L) treatment started at P21. (**B**) Quantification of whole wall proteoglycan/glycosaminoglycan area fraction without or with lonafarnib treatment from P21 reveals a significant improvement with the drug. (**C**) Representative Alizarin Red-stained cross-sections with marked medial calcification were revealed in progeria mice by the dark red color, less with lonafarnib treatment from P21 to P168 (GG + L) relative to untreated progeria. Finally, (**D**) quantitation of calcification, with a trend toward a reduction with lonafarnib treatment. n=5 per group (with three technical replicate sections per specimen), with *, **, and *** denoting statistical significance at p<0.05, p<0.01, and p<0.001, respectively. See also *Figure 4—figure supplement 1* and *Source data 4*.

The online version of this article includes the following figure supplement(s) for figure 4:

**Figure supplement 1.** Similar to *Figure 4* in the main text, but showing additional representative Movat pentachrome (MOV) and Alizarin Red (ALZ) stained sections.

of medial collagen (within 4% of wild-type), a significantly lower accumulation of mural proteoglycans (3.0 fold rather than 3.9 fold greater than wild-type) in progeria (*Figure 4A and B*), and a trend toward lower mural calcification, which manifested primarily between P140 and P168 days in the medial layer of the progeria aorta (*Figure 4C and D*). These multiple histological changes likely contributed to the improved PWV, in part via a numerically lower wall thickness (from 115 to 95 μm in progeria).

## Lonafarnib differentially affects elastic and muscular arteries

Elastic (e.g. the aorta) and muscular (e.g. branch mesenteric) arteries remodel differently in hypertension and natural aging (*Laurent and Boutouyrie, 2015*; *Murtada et al., 2021*). We thus quantified the biomechanical phenotype of the second-order branch mesenteric artery (MA) as a representative muscular artery, again comparing age-matched wild-type and progeria vessels from P42 to P168 (*Figure 5—figure supplement 1*). Both structural (pressure-radius and axial force-stretch) and

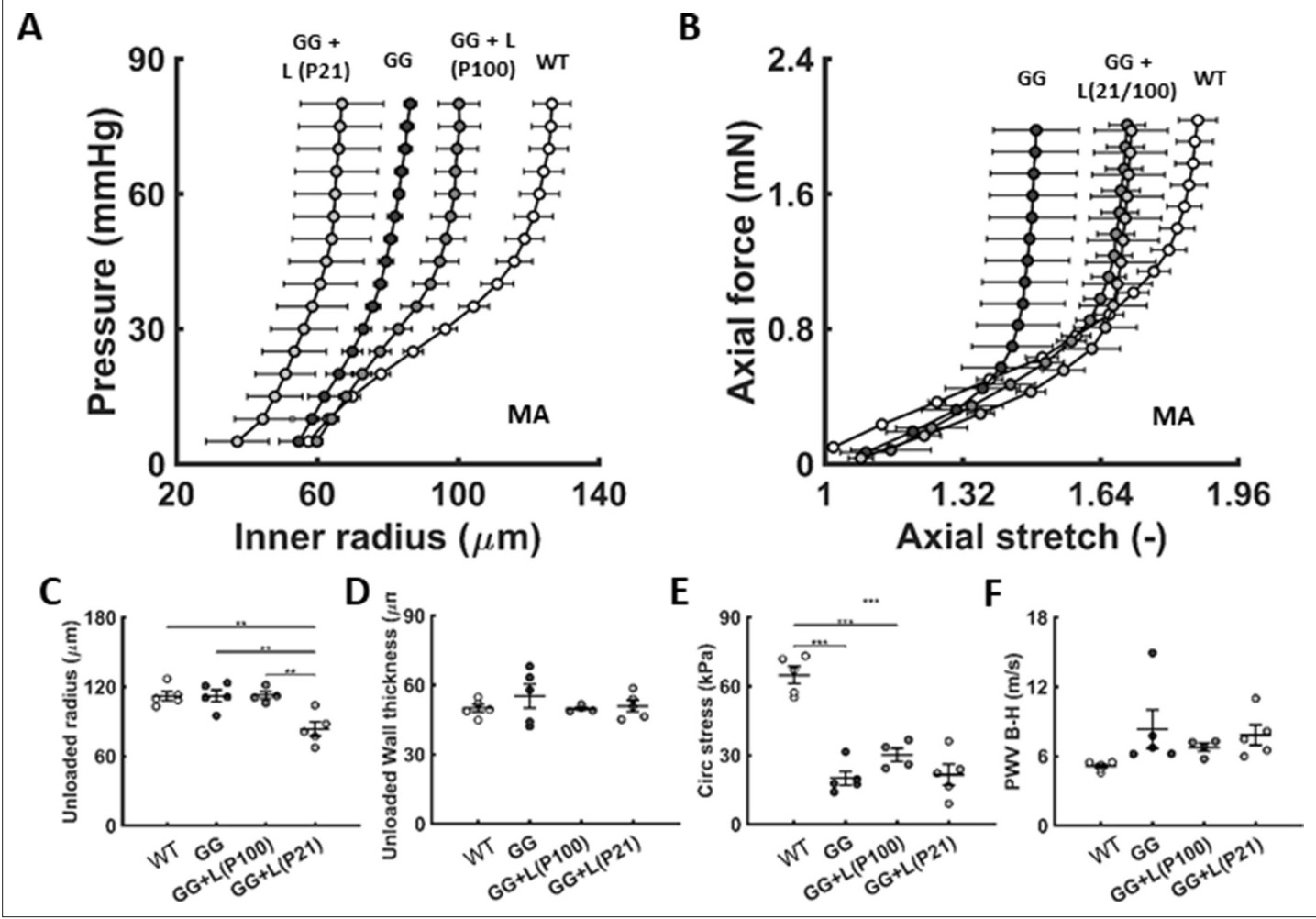

**Figure 5.** Similar to *Figure 2*, except for the second-order branch mesenteric artery in age-matched littermate wild-type (WT) control mice and untreated (GG) and treated (GG + L) progeria mice for lonafarnib administration from either P21 or P100 to P168.n=4–5 per group. (**A-F**) Lonafarnib did not improve passive properties. See also *Figure 5—figure supplement 1* as well as *Source data 1*. Note that we did not assess mesenteric artery properties in the lonafarnib plus rapamycin group given the poor survival outcome.

The online version of this article includes the following figure supplement(s) for figure 5:

**Figure supplement 1.** Similar to *Figure 1*, except for the second-order branch mesenteric artery (MA) from age-matched littermate wild-type (WT) control mice and untreated progeria (GG) mice at postnatal days P42, P100, P140, and P168.

material stiffness were compromised by progeria despite no apparent histological signs of an excessive accumulation of proteoglycans or calcification. Wall thickness was increased, but luminal radius decreased, thus resulting in negligible changes in calculated local PWV. Most dramatic, however, was the progressive decline in vasocontractility from P42 to P168 in the progeria arteries. Treatment with lonafarnib from either P21 or P100 to P168 resulted in a modest shift in the passive pressure-radius behavior relative to untreated but without significant effects on passive geometry and properties (*Figure 5A–F*). Importantly, however, when given from P100 and especially from P21 to P168, lonafarnib largely prevented the decline in vasoconstrictive and vasodilatory capacity seen in progeria mesenteric arteries that were untreated (*Figure 3C and D*). This effect is expected to result in an increased vasoactive capacity and ability to regulate lumen size and blood flow in this muscular artery.

## Lonafarnib improves LV diastolic function and aortic PWV but not body mass

Given the improvements in vascular structure and function when lonafarnib was given from the time of weaning, we focused our echocardiography on this treatment arm. All cardiac measurements

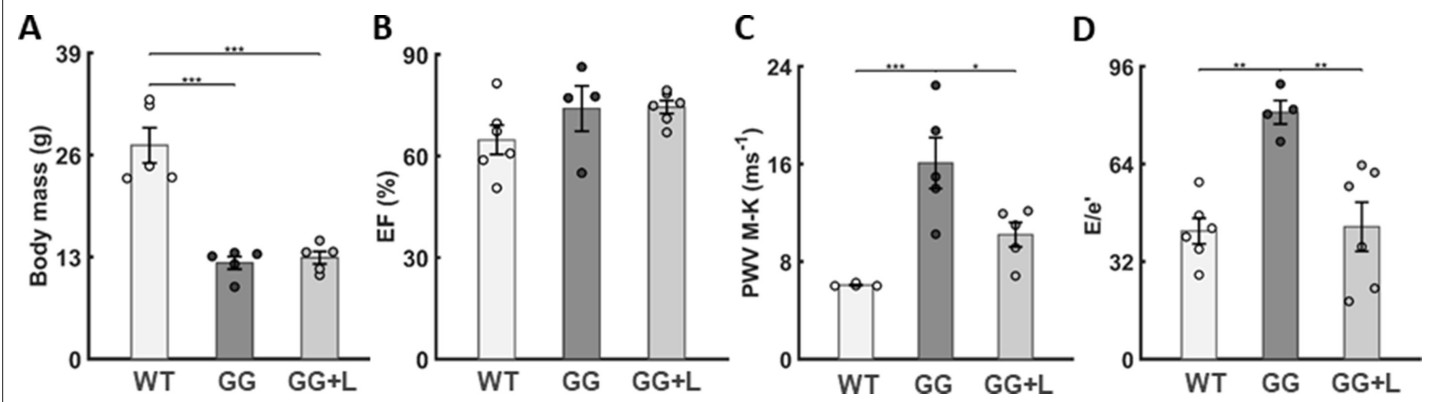

**Figure 6.** Lonafarnib improves pulse wave velocity (PWV) and left-ventricular diastolic function but not somatic growth. Focusing on progeria mice (GG) treated with lonafarnib (GG + L) from P21, (**A**) there was no improvement in body mass measured at P168 despite 100% survival to 168 of the six mice so treated (noting that 1 of the treated mice was found dead on P169 following anesthesia, echocardiology, and successful recovery on P168). (**B**) There was no decline in left ventricular ejection fraction (EF) in the untreated progeria mice and no improvement with lonafarnib. (**C**) Conversely, lonafarnib significantly improved pulse wave velocity, as computed using the Moens-Korteweg equation (PWV M-K), and (**D**) similarly improved left-ventricular diastolic function, as computed by $E/e'$. There was similarly an improvement in cardiac output ($CO$ as detailed in the text). See **Source data 5** for all in vivo cardiovascular measurements (n=4–6 per group), with *, **, and *** denoting statistical significance at p<0.05, p<0.01, and p<0.001, respectively. See also **Figure 6—figure supplement 1A** and **Source data 1–5**.

The online version of this article includes the following figure supplement(s) for figure 6:

**Figure supplement 1.** Additional findings for rapamycin both alone and in combination with lonafarnib.

were compared at 168 days of age for the three primary age-matched study groups: WT, untreated progeria, and progeria treated with lonafarnib from P21 to P168. Noting the lack of improvement in body mass (**Figure 6A**) despite lonafarnib-improved survival (**Figure 2B**), most dimensioned cardiac metrics, such as LV chamber volumes and stroke volume (SV), remained lower in both untreated and lonafarnib-treated progeria mice as expected allometrically based on their lower body mass (**Murtada et al., 2020**). Non-dimensioned metrics such as fractional shortening (FS) and ejection fraction (EF) remained similar across all three groups (WT, untreated, and treated progeria mice) at 168 days of age (**Figure 6B**), suggesting no loss of systolic function (and by inference, no change with treatment). Importantly, consistent with the improved PWV (calculated independently using two different methods; **Figures 2F and 6C**), a key measure of diastolic function (E/e') related to LV filling velocity and mitral motions was preserved in the progeria mice with lonafarnib treatment from P21 to P168 (~43 with treatment vs. 42 for wild-type, both relative to ~81 for untreated progeria mice; **Figure 6D**), consistent with a trend toward an improved cardiac output (from 13 mL/min in progeria to 23 mL/min with treatment, both relative to 20 mL/min for wild-type controls), which likely contributed, in part, to the increased survival.

## Combined lonafarnib and rapamycin did not improve outcomes further

Finally, we examined possible benefits afforded by a combination therapy, daily intraperitoneal injections of the mTOR inhibitor rapamycin from P100 to P168 in progeria mice that received lonafarnib daily from P21 to P168 via the soft chow (**Figure 2A**). Most importantly, there was no improvement in survival relative to the untreated progeria mice (**Figure 2B**), which is to say that combination therapy did not confer the increase in survival achieved with lonafarnib monotherapy. Multiple biomechanical metrics of aortic and cardiac function were contrasted across these groups, which among other findings revealed that combination therapy did not improve PWV (**Figure 2F**), generally consistent with the lack of improvement in intrinsic aortic properties (**Figure 2E and G**, **Figure 2—figure supplement 1**). Finally, there was also no improvement in weight gain (**Figure 6—figure supplement 1A**), no change in left ventricular ejection fraction (**Figure 6—figure supplement 1B**), and no improvement in left ventricular cardiac output (**Figure 6—figure supplement 1C**), which was in stark contrast to the aforementioned improvement in cardiac output achieved with lonafarnib monotherapy. Given these findings, it was surprising that diastolic function (measured in terms of E/e') was similar between the lonafarnib monotherapy and combination therapy (**Figure 6—figure supplement 1D**). Pilot studies

using a rapamycin monotherapy from P100 to P168 also failed to show a survival benefit (*Figure 6— figure supplement 1E*) and were not pursued further. Finally, given the lack of a survival benefit, the effects of this combination treatment were not assessed in the mesenteric artery, which presented with a milder progeria phenotype.

## Discussion

Prior studies have documented marked changes in arterial structure in HGPS, including loss of vascular smooth muscle cells, accumulation of proteoglycans, and increased fibrillar collagen in the adventitia, both in patients (*Stehbens et al., 2001*; *Olive et al., 2010*) and in mouse models (*Varga et al., 2006*; *Capell et al., 2008*; *Kim et al., 2018*). Vascular calcification has also been reported in patients and aged $Lmna^{G609G/+}$ mice (*Villa-Bellosta et al., 2013*) as well as in $Lmna^{G609G/G609G}$ mice crossed with $Apoe^{-/-}$ mice (*Hamczyk et al., 2018*). We recently showed in $Lmna^{G609G/G609G}$ mice that the marked intramural accumulation of proteoglycans in the aortic wall was a driving factor for increasing PWV in the absence of calcification at 140 days of age (*Murtada et al., 2020*). Here, we demonstrate a further rapid increase in PWV in the perimorbid period in these mice (to ~16 m/s at P168) that is driven in part by extensive medial calcification in the aorta, as observed in some patients. Furthermore, we found a continued decline of smooth muscle contractile capacity to P168, consistent with findings at P140 (*Murtada et al., 2020*) as well as with reports of reduced responsiveness to vasodilators (*Varga et al., 2006*).

It has been suggested that $Lmna^{G609G/G609G}$ mice die from starvation and cachexia, with a high-fat diet improving survival in these mice by ~74% from a mean of 111 days to 193 days (*Kreienkamp et al., 2018*). Given the potential role of atherosclerosis in the cardiovascular pathology, however, we provided a normal gel-based diet from weaning that facilitated consumption and helped to improve survival to a maximum of ~180 days, with 53% survival at 168 days. This gel-based diet is a nutritionally fortified dietary supplement that combines hydration and nutrition, but with equivalent levels of fat as regular chow. We submit that this increased survival without the added complications of a high-fat diet allowed time for a severe cardiovascular phenotype to manifest that more closely mimics that reported in HGPS patients. Thus, we were able to study the effects of drug intervention on certain key characteristics of severe cardiovascular disease.

HGPS is a mechano-sensitive condition (*Kirby and Lammerding, 2018*), with stiff tissues and organs most affected. The normal adult murine aorta is highly stressed by hemodynamic loads and it normally expresses high levels of lamin A (*Kim et al., 2018*). Consistent with this observation, we found the progeroid phenotype to be much more severe in the descending thoracic aorta (with passive intramural stresses >200 kPa in normalcy) than in a branch mesenteric artery (with stresses <65 kPa in normalcy). Reduced contractile responses were marked, however, in both the aorta and mesenteric artery. Lonafarnib treatment resulted in only modest long-term improvements in passive stiffness and geometry of the aorta, yet these effects worked together to reduce PWV significantly (by 36.5%, from 16.2 m/s for the untreated progeria mice to 10.3 m/s in progeria mice treated from P21 to P168), likely due to the observed reduction in mural proteoglycans and calcification and preservation of more smooth muscle despite not rescuing the contractile phenotype (noting that aortic smooth muscle cells are primarily responsible for establishing and maintaining the extracellular matrix of the medial layer, not vasoregulating the luminal diameter). It was previously shown that high-dose tipifarnib also reduces proteoglycan accumulation in the aorta (*Capell et al., 2008*) though without a detailed assessment of functional consequences. By contrast, lonafarnib had modest effects on the passive properties of the mesenteric artery but largely preserved its vasoconstrictive and hence endothelial-derived vasodilatory potential. This improved vasoactive capacity appears to have led to a modest reduction in mesenteric artery caliber under passive ex vivo conditions. It is well known that arterioles in the systemic vasculature tend to inward remodel in hypertension (due in part to their myogenic capacity), which may help prevent elevated pressure pulses from propagating into the microcirculation of end organs, thus protecting these organs (*Boutouyrie et al., 2021*). Indeed, we previously found a similar inward remodeling with increased vasoconstrictive capacity in the mesenteric artery in a mouse model of induced hypertension (*Murtada et al., 2021*). Although we did not assess coronary arteries or the coronary microcirculation, an increased vasoregulatory capacity of muscular arteries due to lonafarnib treatment could have combined with the improved central hemodynamics to improve LV diastolic function. In line with this, recent studies have suggested that heart

failure with preserved ejection fraction, which is characterized by the dysfunctional filling of the left ventricle during diastole, associates with coronary artery disease (*Rush et al., 2021*). There is, therefore, a need to investigate further the potential differential effects of lonafarnib on arteries within the many different vascular beds, particularly in end organs, noting again that the normal levels of hemodynamically-induced stresses differ along the vascular tree.

Although lonafarnib increases lifespan in children with HGPS, the disease remains progressive and patients still die early. Hence, multiple studies have considered combination or other therapies. One trial concluded, however, that there was no further cardiovascular benefit of adding the prenylation inhibitors pravastatin and zoledronic acid (a bisphosphonate) to lonafarnib therapy (*Gordon et al., 2016*). Improvement in survival was similarly unremarkable for these combination therapies in $Lmna^{G608G/G608G}$ mice despite improvements in the musculoskeletal phenotype (*Cubria et al., 2020*). Given that mTOR inhibitors, namely rapamycin (*Cao et al., 2011*) and everolimus (*DuBose et al., 2018*), have shown some promise in vitro in treating progeria cells and genetic reduction of mTOR in $Lmna^{G608G/G608G}$ progeria mice extended lifespan ~30% (*Cabral et al., 2021*), we also sought to examine functional effects of lonafarnib + rapamycin versus lonafarnib and rapamycin alone. Rapamycin was given from P100 to P168 via daily intraperitoneal (IP) injections at 2 mg per kg of body mass, consistent with a dosing regime shown to be effective in the treatment of aortopathies in mice (*Li et al., 2014*). We did not find any benefit of the combination therapy when assessing large artery structure or function but instead found early mortality similar to that in untreated progeria mice. Similar findings emerged for rapamycin treatment alone. Given the frailty of these mice, one cannot exclude a detrimental consequence of daily handling and IP injections leading to increased emotional and physical stress, though $Lmna^{G609G/G609G}$ mice have been reported to tolerate two IP injections per week for five weeks (*Lee et al., 2016*). Without any clear benefit, our lonafarnib plus rapamycin study was terminated though other methods of drug administration and other concentrations and durations should be considered, particularly given the aforementioned promising results of genetic reductions in mTOR signaling, though germline manipulations also affect tissues developmentally.

Efforts by others are underway to use genetic manipulations to rescue the progeria phenotype, with one study reporting a 26% increase (*Santiago-Fernández et al., 2019*) and another a 51% increase (*Beyret et al., 2019*) in median survival rate via disruption of $Lmna$ and progerin transcription. Although these studies provide excellent proof-of-principle, genetic editing occurred primarily in the liver, not in the cardiovascular system, and there was no detailed assessment of potential changes in cardiovascular structure or function. Over 90% of children afflicted with HGPS exhibit the c.1824C>T, p.G608G mutation. Targeted anti-sense therapies in $Lmna^{G608G/+}$ and $Lmna^{G608G/G608G}$ mice have extended lifespans from 33 to 62%, from ~220 days to ~330 days (*Erdos et al., 2021*; *Puttaraju et al., 2021*) in these mice, which have a less severe phenotype than $Lmna^{G609G/G609G}$ mice. Most dramatically, gene editing in the $Lmna^{G608G/G608G}$ mouse extended its lifespan more than twofold, from 215 to 510 days of age (*Koblan et al., 2021*). There is, therefore, significant promise for both genetic editing and RNA-based treatment strategies.

Nevertheless, until HGPS clinical trials establish safety and efficacy for genetics-based treatments, pharmacotherapy will likely remain a mainstay in patient care. Importantly herein, daily treatment with lonafarnib from both P21 (weaning) and P100 (maturity) yielded 100% survival of $Lmna^{G609G/G609G}$ mice to the scheduled time of anesthesia and echocardiography at 168 days of age, which can be contrasted with prior reports of extended 50% survival in these same mice to 133 days (with progerinin treatment; *Kang et al., 2021*), 140 days (with JH4 treatment; *Lee et al., 2016*), 161 days (with genetic manipulation; *Santiago-Fernández et al., 2019*), 173 days (with isoprenylcysteine carboxymethyltransferase; *Marcos-Ramiro et al., 2021*), and 177 days (with genetic manipulation; *Beyret et al., 2019*). Although survival was similar between our two lonafarnib cohorts, vascular and cardiac protection was slightly better when treatment was started at weaning. Recalling that LV diastolic dysfunction was the most prevalent cardiovascular abnormality observed in a recent clinical assessment of HGPS patients (*Prakash et al., 2018*), we submit that the increased survival in our $Lmna^{G609G/G609G}$ mice having daily lonafarnib monotherapy likely resulted from a combination of multiple modest improvements in large artery composition and function that yet worked together to significantly improve both central artery hemodynamics (PWV) and LV diastolic function, with improved small artery function also likely contributing by attenuating the propagation of pulse pressure waves into end organs. Importantly, these $Lmna^{G609G/G609G}$ mice, which with lifespan extension via improved feeding and care,

otherwise exhibited a particularly severe aortic phenotype in the perimorbid period reminiscent of the phenotype in children. The present findings thus confirm promise and encourage the use of lonafarnib clinically as the field continues to work to develop a definitive cure (progeriaresearch.org).

# Materials and methods

## Animals

All live animal procedures were approved by the Yale University Institutional Animal Care and Use Committee (IACUC: 2021–11508). $Lmna^{G609G/G609G}$ mice were originally developed by *Osorio et al., 2011*. Herein, mice were generated by breeding $Lmna^{G609G/+}$ mice to obtain homozygous wild-type ($Lmna^{+/+}$=WT) and progeria ($Lmna^{G609G/G609G}$, denoted herein as GG in figures) mice, all having a C57BL/6 genetic background. All mice were fed with the same soft gel-based chow (Clear $H_2O$, DietGel 76 A) placed on the floor of the cage starting at weaning, that is, postnatal day P21. Progeria mice were randomly assigned to treatment versus no-treatment groups. A total of 26 mixed-sex wild-type mice and 68 progeria mice were scheduled for study at P168, noting that we previously found that cardio-vascular effects of progeria were statistically independent of sex at least up to P140 (*Murtada et al., 2020*). Following in vivo collection of echocardiographic data under isoflurane anesthesia, the mice were euthanized using an intraperitoneal injection of Beuthanasia-D either the same (P168) or typically the next (P169) day for tissue harvest, with death confirmed following exsanguination upon removal of the descending thoracic aorta.

## Drug treatment

Lonafarnib was provided by The Progeria Research Foundation Cell and Tissue Bank and given daily at a dose of 450 mg per kg of soft gel-based chow (a dose recommended by the Foundation, noting that healthy adult mice typically eat an equivalent of about 10% of their body mass per day) from either postnatal day P21 or P100 to the study end-point of P168/P169. Rapamycin (Millipore Sigma), at a dose of 2 mg per kg of body mass, was given daily via IP injections from P100 to the study end-point both alone and in combination with a P21-P168 lonafarnib-treated cohort. Additional studies should consider other doses, treatment periods, and ages at treatment initiation, but were beyond the present scope given the limited availability of materials.

## Cardiac function testing

Following standard procedures (*Ferruzzi et al., 2018*; *Murtada et al., 2020*), mice were anesthetized with isoflurane at 168 days of age, and data were collected using a Vevo 2100 ultrasound system (Visualsonic, Toronto, Canada) using a linear array probe (MS550D, 22–55 MHz). Systolic and diastolic function of the left ventricle (LV) were quantified using B-Mode imaging to provide parasternal long axis (LAX) and short axis (SAX) views, with M-Mode imaging in both planes used to track the temporal evolution of cavity diameter at the level of the papillary muscles to measure LV systolic function. B-Mode images of the LV outflow tract diameter and pulsed-wave Doppler images of blood velocity patterns across the aortic valve estimated aortic valve area as a check for valve stenosis. LV diastolic function was monitored using an apical four-chamber view, which was achieved by aligning the ultra-sound beam with the cardiac long axis. Color Doppler imaging located the mitral valve, while pulsed-wave Doppler measured mitral inflow velocity. Finally, Doppler tissue imaging from the lateral wall and interventricular septum measured the velocities of tissue motion.

Echocardiographic data were processed using Visualsonic data analysis software and standard methods. Global parameters of the LV function were measured in technical triplicates from individual parasternal LAX and SAX views, and final parameters were obtained as averages from both views. Herein, LV systolic function is characterized by parameters such as cardiac output (CO) and ejection fraction (EF), while LV diastolic functional parameters include peak early (E) filling velocity and peak atrial (A) filling velocity, and their ratio (E/A), as well as the E-wave deceleration time (DT), and tissue velocity (e').

Although we routinely measure invasive blood pressures at the conclusion of echocardiography in adult mice (via a carotid cut-down and Millar catheter while the mouse remains anesthetized), our attempts in the fragile low-weight untreated progeria mice were inconsistent and generally

unsuccessful, thus preventing any reliable comparison of invasive blood pressures across groups. Hence, we do not report any blood pressure.

## Biomechanical testing

Our biomechanical testing and analysis follow validated standardized protocols (*Ferruzzi et al., 2015*; *Murtada et al., 2020*). Briefly, excised vessels (descending thoracic aorta and the second-order branch mesenteric artery) were mounted on custom-drawn glass cannulae and secured with a 6–0 suture. They were then placed within a custom computer-controlled biaxial device for biomechanical testing (see *Gleason et al., 2004*, which includes line drawings and a parts list for the device) and immersed in a heated (37 °C) and oxygenated (95% $O_2$, 5% $CO_2$ to maintain a physiological pH) Krebs-Ringer bicarbonate buffered solution containing 2.5 mM $CaCl_2$. The vessels were then subjected to a series of isobaric (distending pressure of 90 mmHg for the aorta, 60 mmHg for the mesenteric artery) - axially isometric (fixed specimen-specific in vivo axial stretch) protocols wherein they were contracted with 100 mM KCl, relaxed (KCl washed out), contracted with 1 µM phenylephrine (PE), then dilated with 10 µM acetylcholine (ACh; without wash-out), an endothelial cell-dependent stimulant of nitric oxide production. Of course, Ach-stimulated vasodilation is possible only following pre-constriction in this ex vivo protocol, hence lack of Ach-stimulated vasodilation of a vessel that cannot contract does not provide information on endothelial cell function.

Upon completion of these active protocols, the normal Krebs solution was washed out and replaced with a $Ca^{2+}$-free Krebs solution to ensure a sustained passive behavior while maintaining heating and oxygenation. Vessels were then preconditioned via four cycles of pressurization (between 10 and 140 mmHg for the aorta, between 10 and 90 mmHg for the mesenteric artery) while held fixed at their individual in vivo axial stretch. Finally, specimens were exposed to seven cyclic protocols: three pressure-diameter (*P-d*) protocols, with luminal pressure cycled between 10 and either 140 mmHg (aorta) or 90 mmHg (mesenteric) while axial stretch was maintained fixed at either the specimen-specific in vivo value or ±5% of this value, plus four axial force-length (*f-l*) tests, with force cycled between 0 and a force equal to the maximum value measured during the pressurization test at 5% above the in vivo axial stretch value, while the luminal pressure was maintained fixed at either {10, 60, 100, or 140 mmHg} for the aorta or {10, 30, 60, 90} for the mesenteric artery. Pressure, axial force, outer diameter, and axial length were recorded online for all seven of these protocols and the over 2800 data points per vessel were used for subsequent data analysis (using the last unloading curve of each of the seven protocols, which provides information on elastically stored energy that is available to work on the distending fluid). Prior studies have revealed robust fixed-point estimates of constitutive parameters (see below) based on such data.

## Passive descriptors

Five key biomechanical metrics define well the passive mechanical phenotype of the aorta: mean circumferential and axial wall stress, circumferential and axial material stiffness linearized about a physiologic value of pressure and axial stretch, and the elastically stored energy. Each of these metrics is calculated easily given best-fit values of the eight model parameters within a nonlinear stored energy function, herein taken as (*Ferruzzi et al., 2015*; *Murtada et al., 2020*)

$$W = \frac{c}{2}\left(I_C - 3\right) + \sum_{i=1}^{4} \frac{c_1^i}{4c_2^i} \left\{ exp\left[ c_2^i \left( IV_C^i - 1 \right)^2 \right] - 1 \right\}, \tag{1}$$

where $c$ (dimension of kPa), $c_1^i$ (dimension of kPa), and $c_2^i$ (dimensionless), with $i = 1, 2, 3, 4$, are model parameters. $I_C = C : I$ and $IV_C^i = C : M^i \otimes M^i$ are coordinate invariant measures of deformation, with $I$ the identity tensor, $C = F^T F$ the right Cauchy-Green tensor, and $F$ the deformation gradient tensor (mapping from the near traction-free configuration to any loaded configuration, which is pressurized and axially stretched herein) and superscript $T$ the transpose of the tensor; $det F = 1$ because of assumed incompressibility. The direction of the $i^{th}$ family of fibers is given by the unit vector $M^i = \left[ 0, sin\alpha_0^i, cos\alpha_0^i \right]$, with angle $\alpha_0^i$ computed with respect to the axial direction in the traction-free reference configuration. Based on prior microstructural observations, and the yet unquantified effects of cross-links and physical entanglements amongst the multiple families of fibers, we included contributions of axial ($\alpha_0^1 = 0$), circumferential ($\alpha_0^2 = \pi/2$), and two symmetric diagonal families of fibers

($\alpha_0^{3,4} = \pm\alpha_0$) to capture phenomenologically the complex biaxial material behavior; this relation has been validated independently by multiple groups.

Theoretical values of the applied loads were computed from components of Cauchy stress by solving standard global equilibrium equations in radial and axial directions (*Humphrey, 2002*), given the assumption of axisymmetry. Best-fit values of the eight model parameters ($c, c_1^1, c_2^1, c_1^2, c_2^2, c_1^{3,4}, c_2^{3,4}, \alpha_0^{3,4}$) were estimated via nonlinear regression (Levenberg-Marquardt) to minimize the sum-of-the-squared differences between experimentally-measured and theoretically-predicted values of luminal pressure and axial force, each normalized by average experimental measures (*Ferruzzi et al., 2015*). Best-fit values were defined by the minimum value of the objective function based on three randomly selected sets of initial guesses. Estimated parameters (constrained to be non-negative) were used to compute stress, material stiffness, and stored energy at any configuration. For example, components of the stiffness tensor ($C_{ijkl}$), linearized about a configuration defined by the distending pressure and in vivo value of axial stretch, were computed as

$$C_{ijkl} = 2\delta_{ik}F_{lA}^o F_{jB}^o \frac{\partial W}{\partial C_{AB}} + 2\delta_{jk}F_{iA}^o F_{lB}^o \frac{\partial W}{\partial C_{AB}} + 4F_{iA}^o F_{jB}^o F_{kP}^o F_{lQ}^o \frac{\partial^2 W}{\partial C_{AB}\partial C_{PQ}}|_{C^o} \tag{2}$$

where $\delta_{ij}$ are components of $I$, $F^o$ is the deformation gradient tensor between the chosen reference configuration and a finitely deformed in vivo relevant configuration, and $C^o$ is the corresponding right Cauchy-Green tensor. Local aortic PWV was calculated using both the Moens-Korteweg (based on material stiffness and wall geometry) and Bramwell-Hill (based on overall wall compliance) equations, which yielded similar results, thus providing an internal validation of rigor. The former is often preferred for it better delineates contributors to this important metric: circumferential material stiffness, wall thickness, and luminal radius.

## Histology

Following biomechanical testing, specimens were unloaded and fixed overnight in 10% neutral buffered formalin, then stored in 70% ethanol at 4 °C for histological examination. Fixed samples were dehydrated, embedded in paraffin, sectioned serially (5 μm thickness), and stained with Movat Pentachrome (showing elastic fibers in black, cytoplasm in red, glycosaminoglycans in blue, collagen in gray-yellow) or Alizarin Red (showing calcium in dark red) for standard histology. Detailed analyses were performed on a minimum of three biomechanically representative vessels per group, with three technical replicates (sections) per vessel. Histological images were acquired on an Olympus BX/51 microscope (under bright light imaging) using an Olympus DP70 digital camera (CellSens Dimension) and a 20 x magnification objective. Custom MATLAB scripts extracted layer-specific cross-sectional areas and calculated positively stained pixels and area fractions (https://github.com/yale-cbl/histological-analysis; copy archived at *Yale Continuum Biomechanics Laboratory, 2023*).

## Statistics

All biomechanical data represent biological, not technical, replicates. Importantly, the data shown for all geometric and mechanical vascular metrics came from computer-controlled ex vivo testing that reduced specimen-to-specimen variability relative to standard testing while generating over 2800 data points per sample, from which nonlinear regression yielded best-fit parameters in a validated constitutive relation (*Equation 1*) from which measures of stress, stiffness, energy, PWV, etc. were determined. Based on copious prior studies using similar biomechanical data, one- or two-way analysis of variance (ANOVA) with Bonferroni post hoc testing was used to compare results, with p<0.05 considered significant. All data are presented as mean ± standard error of the mean (SEM).

## Standards and data availability

The study was designed consistent with ARRIVE standards with the exception that mice were allocated to groups sequentially, not randomly. That is, only progeria mice were treated and groups were completed in sequence, lonafarnib first, then rapamycin and lonafarnib, then rapamycin alone. Numerical values of all cardiovascular metrics are found in Source Data files, and custom software for histological analysis is found at: https://github.com/yale-cbl/histological-analysis (copy archived at *Yale Continuum Biomechanics Laboratory, 2023*).

## Acknowledgements

This work was supported, in part, by grants from the US National Institutes of Health (R01 HL105297, R21 AG067347) and Inozyme Pharma, Inc (to DTB). We also thank The Progeria Research Foundation (progeriaresearch.org) for the generous donation of lonafarnib. This work was presented on November 4, 2022 during the 11th International Scientific Workshop of the Progeria Research Foundation in Boston.

## Additional information

### Competing interests

Demetrios T Braddock: DTB is an equity holder and receives research and consulting support from Inozyme Pharma, Inc. DTB is also an inventor on patents owned by Yale University that are unrelated to the current study, but can be viewed nonetheless at https://medicine.yale.edu/lab/braddock/intellectual_property/. The other authors declare that no competing interests exist.

### Funding

| Funder | Grant reference number | Author |
| --- | --- | --- |
| National Heart Lung & Blood Institute | R01 HL105297 | Jay D Humphrey |
| National Institute on Aging | R21 AG067347 | Demetrios T Braddock |
| Inozyme Pharma | | Demetrios T Braddock |

The funders had no role in study design, data collection and interpretation, or the decision to submit the work for publication.

### Author contributions

Sae-Il Murtada, Conceptualization, Data curation, Formal analysis, Investigation, Methodology, Writing – original draft, Writing – review and editing; Nicole Mikush, Mo Wang, Pengwei Ren, Yuki Kawamura, Abhay B Ramachandra, Data curation, Formal analysis, Methodology; David S Li, Resources, Data curation, Formal analysis, Funding acquisition, Investigation, Writing – review and editing; Demetrios T Braddock, Resources, Funding acquisition, Investigation, Methodology; George Tellides, Conceptualization, Resources, Investigation, Methodology, Project administration, Writing – review and editing; Leslie B Gordon, Conceptualization, Formal analysis, Supervision, Funding acquisition, Investigation, Methodology, Writing – original draft, Project administration, Writing – review and editing; Jay D Humphrey, Conceptualization, Data curation, Formal analysis, Supervision, Funding acquisition, Investigation, Methodology, Writing – original draft, Project administration, Writing – review and editing

### Author ORCIDs

Yuki Kawamura http://orcid.org/0000-0003-2137-6464
Jay D Humphrey http://orcid.org/0000-0003-1011-2025

### Ethics

All live animal procedures were approved by the Yale University Institutional Animal Care and Use Committee (IACUC: 2021-11508).

### Decision letter and Author response

Decision letter https://doi.org/10.7554/eLife.82728.sa1
Author response https://doi.org/10.7554/eLife.82728.sa2

## Additional files

### Supplementary files

- MDAR checklist
- Source data 1. Passive mechanics.

- Source data 2. Active mechanics.
- Source data 3. Survival information.
- Source data 4. Histological information.
- Source data 5. Echo data.

## Data availability

All numerical data generated or analyzed during this study are included in the manuscript, supplemental materials, and supporting source data (Excel) files.

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
