## [Editor Report]

The authors performed an important study to demonstrate efficacy of lonafarnib, the only approved drug for treating progeria in patients, using a well-established mouse model of the disease. The authors provided convincing evidence that lonafarnib extends lifespan in these mice that is associated with reduced arterial thickening and improved arterial function in an experimental model progeria. This paper is of major interest to scientists and clinicians interested in cardiovascular physiopathology, aging, and progeria in particular.

---

## [Decision Letter]

**Decision letter after peer review:**

Thank you for submitting your article "Lonafarnib improves cardiovascular function and survival in a mouse model of Hutchinson-Gilford Progeria Syndrome" for consideration by *eLife*. Your article has been reviewed by 3 peer reviewers, including Daniel Henrion as Reviewing Editor and Reviewer #3, and the evaluation has been overseen by a Senior Editor.

The reviewers have discussed their reviews with one another, and the Reviewing Editor has drafted this letter to help you prepare a revised submission.

Essential revisions:

1 – The dose of 450 mg/kg of lonafarnib is quite high in the light of the dose of other compounds or even the oral dose of lonafarnib approved for children. This should be discussed, since a possible limitation of the study is whether a dose closer to the clinically used in patients (ie, 7.5 mg/kg) is able to produce cardiovascular improvements in mice.

2 – Rapamycin did not improve the phenotype when added to lonafarnib. The effect of rapamycin alone should be shown, or rapamycin should be removed from this work. Furthermore, the rationale for the use of rapamycin should be better explained in the introduction.

3 – A group with lonafarnib administered to wild type mice is necessary for a better understanding of the effect of lonafarnib on body weight as well as on cardiac and arterial function and structure.

4 – The effect of lonafarnib on proteoglycans accumulation cannot be appreciated from the images shown in the corresponding figures. The technique used for normalization, analysis and the data presentation should be clearly detailed.

5 – The authors should provide more insight on the mechanism by which lonafarnib reduced proteoglycan accumulation.

6 – Considering the importance of the number of vascular smooth muscle cells (VSMCs) in progeria (Cells 2020, 9, 656; Circulation. 2018, 138, 266; *eLife* 2020, 9, e54383) this parameter should be analysed more in depth. The cell composition of the vascular (aorta and mesenteric arteries) should be detailed in GG, WT and treated mice (SMCs and type of SMCs, endothelial cells and adventitial cells including immune cells). In this regard,

7 – Endothelium-dependent dilation should be analysed more in depth. The loss of both SMCs and ECs function suggests a more global effect of the disease on vascular cells. As lonafarnib prevented this loss of cellular function in mesenteric arteries, SMCs and endothelial function should be shown in the aorta as well.

8 – Arterial diameter is reduced in GG mice (aorta and mesenteric artery). As small arteries control tissues perfusion and pressure it is important to provide blood pressure values in the different groups of mice. A measurement of tissue perfusion (laser Doppler of the mouse feet for example) would also add a lot to the study.

9 – In page 9 (lines 257-260) the authors put in context the obtained results regarding survival with other treatments. However, there is one study that should be also incorporated, since 174 days are obtained in 50% survival extension after administration of an ICMT inhibitor (ACS Central Sci. 2021, 7, 8, 1300).

10 – Statistical analysis and number of mice used: The text and the analysis shown in figures do not always match. This should be corrected carefully. See the specific comment by reviewer 2.

11 – All the abbreviations should be defined (see in particular: PWV in page 4; ALZ and MOV in Figure 4).

12 – Number of animals and their age (age at the time of the experiments) must be given in each legend.

13 – The quality of the figure must be improved (for example: 1G and 1H, legend difficult to read)

14 – References should be listed in alphabetical order.

*Reviewer #1 (Recommendations for the authors):*

The objective of this work is to analyze the effect of administration of lonafarnib, the only FDA approved drug for treating progeria, on the main cardiovascular alterations that characterize the disease using a progeria mouse model. In particular, the authors describe that lonafarnib ameliorates aortic phenotype (in terms of PWC, proteoglycan composition and elasticity) and LV diastolic function. The methodology and the experiments carried out are appropriate, and the conclusions of the work are in general well supported by the data. The most significant result is that the reported data provide a molecular rationale in relation with improvements in specific cardiovascular parameters that explain the phenotypic benefits observed upon lonafarnib administration. All in sum, the findings reported in this work are of interest for those working in the area of progeria and other laminopathies and support the use of lonafarnib for treating progeria.

The conclusions of this paper are mostly well supported by data, but some aspects need to be clarified.

– There are some abbreviations that are specific to the area of cardiovascular research. They should be defined for a better understanding of the readership not specialized in the cardiovascular field (for example PWV in page 4; ALZ and MOV in Figure 4).

– The use of a dose of 450 mg/kg of lonafarnib seems quite high in the light of the dose of other compounds or even the oral dose of lonafarnib approved for children. As the authors attempt to support the use of lonafarnib in humans on the basis of the cardiovascular improvement observed in this study in mice, the dose should be discussed. The dose in humans is 150 mg per m2 of body surface. Considering 0,5 m2 of surface per 10 kg of body weight (https://www.ouh.nhs.uk/oxparc/professionals/documents/Body-surfaceareaCCLGChart1.pdf) a dose of 75 mg represent around 7.5 mg/kg. This is very far from the 450 mg/kg indicated in this paper. This should be discussed, since a possible limitation of the study is whether a dose more close to the clinically used in patients (ie, 7.5 mg/kg) is able to produce cardiovascular improvements in mice.

– Since no significant improvement in body mass is observed (line 117, page 4) it should be discussed whether lonafarnib administered to wild type mice affects to body weight.

– The paper indicates that lonafarnib improves aortic composition, in particular lowering accumulation of mural proteoglycans (line 142, page 5, and Figures 4B and 4C). It would be nice to mark/highlight in the corresponding figures the area or the specific zone in the image where these improvements can be appreciated.

– In relationship with the former result, is there any insight on the mechanism by which lonafarnib reduces proteoglycan accumulation?

– Considering the importance of the number of vascular smooth muscle cells (VSMCs) in progeria (Cells 2020, 9, 656; Circulation. 2018, 138, 266; *eLife* 2020, 9, e54383) this parameter should be also analyzed.

– In page 9 (lines 257-260) the authors put in context the obtained results regarding survival with other treatments. However, there is one study that should be also incorporated, since 174 days are obtained in 50% survival extension after administration of an ICMT inhibitor (ACS Central Sci. 2021, 7, 8, 1300). This work should be also cited.

– References are difficult to find, they should be listed in alphabetical order.

*Reviewer #2 (Recommendations for the authors):*

This paper focuses on the effects of the drug Lonafarnib, which is a farnesyltransferase inhibitor used as a therapeutic for patients with Hutchinson-Gilford progeria syndrome (HGPS) an accelerated aging disease characterized by premature death due to cardiovascular problems. Lonafarnib significantly increases lifespan, but it's effects on cardiac function have not been documented. The authors used an established mouse model for HGPS that carry the point mutation of the LMNA gene exhibiting severe cardiovascular disease and shortened life span. Homozygous G609G mice show progressive thoracic aorta structural stiffening and significant reduction in biomechanical function. Lonafarnib significantly improved survival, left ventricular diastolic function and descending thoracic aortic pulse-wave velocity, with lower accumulation of mural proteoglycans and calcification in the aorta. Studies also showed a moderate ability of Lonafarnib to prevent decline in vascoconstrictive and vasodilatory capacities of the muscular-type mesenteric artery. Finally, combined therapy with the MTOR inhibitor rapamycin was not beneficial, and even deleterious regarding the pulse wave velocity of the aorta. The authors conclude that Lonafarnib treatment can significantly improve both arterial structure and function to reduce pulse wave velocity and left ventricular diastolic function.

Strengths

This paper provides important information on Lonafarnib's effects as a therapeutic agent in HGPS. By maintaining the mouse model of HGPS on a specific gel diet, the authors were able to extend the lifespan of the mice, and thus the cardiovascular disease phenotype, without resorting to external parameters such as high-fat diet, which would have created a bias towards atherosclerosis and the metabolic syndrome. The paper is well written with the objectives stated clearly.

Weaknesses

A question that arises is with regards to the combination therapy (Figure 2, Table S1). There is no data provided on the HGPS mice treated with rapamycin alone, thus making it difficult to understand how the authors can determine the reasons underlying the absence or worsening of effects of both Lonafarnib and rapamycin. If this data is available, it should be included. If not, it is questionable if this data as it brings additional value to the paper. It would also be helpful to the reader to present the reasoning behind dual treatment with Ioanafarnib and rapamycin in the introduction.

There are some imprecise statements in the text in which the authors indicate changes in parameters, when in fact, the differences are not statistically significant. For example, lines 123-124 (Table 1) "Most dimensioned cardiac metrics, such as LV chamber volumes and stroke volume (SV), were lower in both untreated and lonafarnib-treated progeria mice…", when in fact it is not indicated as being statistically significant in Table 1. Also, in Figure 2F legend, it is indicated that lonafarnib improved cardioac output, when in the text, it is indicated as a trend toward improved cardiac output (line 129). The text should be reviewed carefully and corrected.

Figure 1. In Figure 1A, it would be helpful to include data on the WT littermates providing a reference to the change in structural stiffness as a function of age.

Figure 4. The authors report significantly decreased proteoglycan accumulation in the group that received lonafarnib. This is not evident based on the images. For the analysis, how did the authors normalize the values obtained, the amount of proteoglycan and calcification was relative to what parameter?

*Reviewer #3 (Recommendations for the authors):*

The authors have investigated the effect of lonafarnib, a farnesyltransferase inhibitor, in a mouse model of progeria. The treatment with lonafarnib extended lifespan in progeria mice, reduced arterial thickening, and improved arterial function in progeria mice. This treatment increases the lifespan of the mice and prevents, at least in part, the thickening, and the loss of contractility of the arteries using appropriate in vivo and in vitro tools. This study brings new information on the pathophysiology of progeria and on the mechanism of lonafarnib which improves arterial wall structure and contractility.

1 – The data shown in figure 1 need to be reinforced. Indeed, in GG mice the aorta becomes translucent whereas its thickness increases to more than 100µm. Such a tremendous change in structure requires a more in-depth analysis. It is important to show not only the degree of calcification (this is shown / alizarin red) but also the location of the cells in the wall (and the type of cells). Is there any smooth muscle left? Which phenotype do they have, and which other cells have eventually infiltrated the vascular wall if any (or the adventitia which is very thick in GG mice). Figure 4 shows a more limited thickening of the aortic wall. Indeed, Movat staining shows a relatively well-structured media and a thicker adventitia. This phenotype suggests a strong inflammatory response of the vascular wall which could explain the loss of contractility in GG mice. This needs to be analyzed more in depth.

2 – Contractility and dilation: The strong reduction in contractility in the aorta and in mesenteric arteries suggests a major SMC defect. In parallel, endothelium-mediated (Ach) dilation is also abolished in mesenteric arteries suggesting that endothelial cells (ECs) are also affected. Ach-dependent dilation should also be measured in the aorta. This loss of SMCs and ECs function suggests a more global effect of the disease on vascular cells. Interestingly, the treatment with lonafarnib prevents this loss of cellular function (and loss of cells?) in the mesenteric arteries. This is not shown in the aorta whereas this information seems essential. Indeed, it's important to have the reactivity data in figure 3 (as in figure 1 G, H and I + Ach).

3 – Arterial diameter: In both the aorta and the mesenteric artery, arterial diameter is severely reduced in GG mice. As small arteries control tissues perfusion and pressure it is important to provide the blood pressure values in the different groups of mice. In the mesenteric arteries the inward remodeling resembles that seen in hypertension as discussed by the authors. The authors should measure tissue perfusion (laser Doppler of the mouse feet for example).

In addition, an histological analysis of the myocardium is necessary to assess whether hypertrophy or excessive dilation occurs in GG mice with arterial thickening.

4 – Lonafarnib did not prevent this inward remodeling whereas it improved contractility and dilation in MA (how about the aorta?). Again, blood pressure and tissue perfusion are necessary to understand the effect of the treatment.

---

## [Author Response]

Essential revisions:1 – The dose of 450 mg/kg of lonafarnib is quite high in the light of the dose of other compounds or even the oral dose of lonafarnib approved for children. This should be discussed, since a possible limitation of the study is whether a dose closer to the clinically used in patients (ie, 7.5 mg/kg) is able to produce cardiovascular improvements in mice.

Thank you for revealing that our description was not clear, probably because of a misplaced comma (now corrected, but also with a clearer description). The dose was 450 mg of lonafarnib per kg of food (not per kg of mouse body mass), now stated explicitly. Normal mice can eat up to 10% of their body mass per day, though the continued loss of body mass in progeria mice suggests lower levels of daily consumption. At the maximum expected level (10% of body mass per day), each progeria mouse could ingest 0.675 mg of lonafarnib per day, which would translate into 45 mg/kg per mouse maximum. At half that level, this would translate into 22.5 mg/kg, much closer to that stated above for children (7.5 mg/kg). This is now stated clearly.

Note, in addition, that we received the lonafarnib as an in-kind gift from the Progeria Research Foundation (PRF) under formal agreement and we used their recommended dosage, per their request (now noted). PRF is the international leader in directing progeria research and their opinion was valued, their request was respected.

Finally, we also now note (in Discussion) that additional doses as well as periods of treatment and ages of initiation of treatment would be important to evaluate in the future. As an example, we now include a lonafarnib treatment arm during which drug was given from P100 to P168 (in contrast to the originally and still reported P21 to P168, which yielded better outcomes). For completeness, we now report and compare both of these treatment arms.

2 – Rapamycin did not improve the phenotype when added to lonafarnib. The effect of rapamycin alone should be shown, or rapamycin should be removed from this work. Furthermore, the rationale for the use of rapamycin should be better explained in the introduction.

As the Reviewer may know, lonafarnib has shown benefits in progeria children but there are ongoing studies to try to identify combination therapies that improve outcomes further. One such study combines lonafarnib and a mTOR inhibitor. It was our goal, therefore, to test a combination therapy as well, noting that it will be unlikely that any future clinical trial will not include lonafarnib given the positive results to date. Indeed, within the formal agreement we received from PRF they state: “…clinical trials testing new potential medications for HGPS will involve at least one combination treatment arm, administering lonafarnib plus a new drug of interest. Therefore PRF suggests that researchers consider similar preclinical studies, with one arm testing combination treatment with lonafarnib plus new compound of interest.” Again, we respected this request and focused on lonafarnib + rapamycin. Because of the poor outcome, however, we obviously evaluated rapamycin alone – the results were similarly poor.

In an attempt to simplify the message of the initial paper, we had left out the set of data using rapamycin alone from P100 to P168. As requested, these data are now included as a survival curve. Given the poor survival outcomes, however, we did not perform complete cardiovascular assessments (indeed could not since most of the mice died prior to the study endpoint). Now noted.

3 – A group with lonafarnib administered to wild type mice is necessary for a better understanding of the effect of lonafarnib on body weight as well as on cardiac and arterial function and structure.

Here we disagree. There was no difference in body mass between the lonafarnib and non-lonafarnib treated progeria mice and thus no expectation that lonafarnib led to any further decrease in body mass. We did not consider lonafarnib treatment of wild-type mice.

4 – The effect of lonafarnib on proteoglycans accumulation cannot be appreciated from the images shown in the corresponding figures. The technique used for normalization, analysis and the data presentation should be clearly detailed.

Thank you for this important comment, we agree. We have since re-confirmed our method of quantification and had a second investigator not involved in the original study re-analyze the entire data set de novo, resulting in a new Figure with updated values, which are not different from those reported previously hence the prior conclusions remain. We feel that the visual appreciation of the effects of progeria and its treatment is much improved in this new figure.

5 – The authors should provide more insight on the mechanism by which lonafarnib reduced proteoglycan accumulation.

Despite an increasing data base on vascular changes in progeria, there is currently no well accepted mechanism by which proteoglycans accumulate in the large arteries in progeria (or why they don’t accumulate so dramatically in small arteries) and thus it is not possible to determine how lonafarnib affects this yet unknown mechanism. Our focus, rather, was on potential systemic effects on cardiac and large artery function, which have direct clinical relevance. Albeit not emphasized as much as it should have been, we now discuss the need for more attention to these mechanisms.

6 – Considering the importance of the number of vascular smooth muscle cells (VSMCs) in progeria (Cells 2020, 9, 656; Circulation. 2018, 138, 266; eLife 2020, 9, e54383) this parameter should be analysed more in depth. The cell composition of the vascular (aorta and mesenteric arteries) should be detailed in GG, WT and treated mice (SMCs and type of SMCs, endothelial cells and adventitial cells including immune cells). In this regard,

We agree that extreme loss of vascular SMCs has been one of the most consistent observations in progeria studies of both patients and murine models. We now include in our new Figure (on medial cell density) data that are consistent with this observation: there was a measured 79% reduction in SMC area with progeria at P168 in the aorta, which was slightly less (67% reduction) with lonafarnib. We also show in a revised figure, however, that there was no recovery of SMC function in the aorta with lonafarnib treatment (previously simply stated, not shown in a figure) despite less SMC loss whereas there was significant recovery in SMC function in small arteries. Thank you for this comment for these are important clarifications.

Yes, it would be interesting to assess other cell types, but it appears (and largely confirmed at the 11^th^ International Progeria Workshop at Boston in Nov 2022) that SMCs are most important – it was their function that we focused on and thus evaluated, noting that we did assess endothelial function via Ach stimulation (see next point below).

7 – Endothelium-dependent dilation should be analysed more in depth. The loss of both SMCs and ECs function suggests a more global effect of the disease on vascular cells. As lonafarnib prevented this loss of cellular function in mesenteric arteries, SMCs and endothelial function should be shown in the aorta as well.

Traditional in vitro studies of endothelium-dependent dilatation require that the vessels be preconstricted (i.e., a vessel cannot dilate unless previously contracted). Thus, in the absence of vasoconstrictive capacity, one cannot assess effects of ECs on dilatation. We have revised / expanded our Figures contrasting SMC contraction / EC-dependent dilatation in large and small arteries, before and after lonafarnib treatment, while noting the experimental caveat regarding the inability of assessing functional vasodilation in the absence of prior vasoconstriction.

8 – Arterial diameter is reduced in GG mice (aorta and mesenteric artery). As small arteries control tissues perfusion and pressure it is important to provide blood pressure values in the different groups of mice. A measurement of tissue perfusion (laser Doppler of the mouse feet for example) would also add a lot to the study.

First, regarding the larger vessels, we previously found that the smaller diameters largely scale allometrically with the reduced body mass (as would be expected; Murtada et al., 2020), which is to say that smaller appears to be physiologically relevant – emphasizing the importance of not just sex- and age-sensitive analyses, but body mass-sensitive as well.

Regarding the smaller vessels, we agree that increased contractile function can alter tissue perfusion as well as affect the potential propagation of the pulse wave into distal tissues and organs. We routinely obtain blood pressures (non-invasive and invasive, using Millar catheters for mice) in healthy adult mice and attempted to obtain such pressures in the progeria mice. Given their small size and fragility, however, we were not able to collect consistent measurements of central blood pressure despite many attempts. This is now noted in the revision.

9 – In page 9 (lines 257-260) the authors put in context the obtained results regarding survival with other treatments. However, there is one study that should be also incorporated, since 174 days are obtained in 50% survival extension after administration of an ICMT inhibitor (ACS Central Sci. 2021, 7, 8, 1300).

Thank you. This omission has been corrected.

10 – Statistical analysis and number of mice used: The text and the analysis shown in figures do not always match. This should be corrected carefully. See the specific comment by reviewer 2.

We have checked, corrected, and expanded the study as needed. Numbers of mice in each group are shown correctly in each table.

11 – All the abbreviations should be defined (see in particular: PWV in page 4; ALZ and MOV in Figure 4).

Agreed and corrected.

12 – Number of animals and their age (age at the time of the experiments) must be given in each legend.

Agreed and corrected.

13 – The quality of the figure must be improved (for example: 1G and 1H, legend difficult to read)

Agreed and corrected.

14 – References should be listed in alphabetical order.

Corrected.

Reviewer #1 (Recommendations for the authors):The objective of this work is to analyze the effect of administration of lonafarnib, the only FDA approved drug for treating progeria, on the main cardiovascular alterations that characterize the disease using a progeria mouse model. In particular, the authors describe that lonafarnib ameliorates aortic phenotype (in terms of PWC, proteoglycan composition and elasticity) and LV diastolic function. The methodology and the experiments carried out are appropriate, and the conclusions of the work are in general well supported by the data. The most significant result is that the reported data provide a molecular rationale in relation with improvements in specific cardiovascular parameters that explain the phenotypic benefits observed upon lonafarnib administration. All in sum, the findings reported in this work are of interest for those working in the area of progeria and other laminopathies and support the use of lonafarnib for treating progeria.The conclusions of this paper are mostly well supported by data, but some aspects need to be clarified.– There are some abbreviations that are specific to the area of cardiovascular research. They should be defined for a better understanding of the readership not specialized in the cardiovascular field (for example PWV in page 4; ALZ and MOV in Figure 4).

Agreed and corrected.

– The use of a dose of 450 mg/kg of lonafarnib seems quite high in the light of the dose of other compounds or even the oral dose of lonafarnib approved for children. As the authors attempt to support the use of lonafarnib in humans on the basis of the cardiovascular improvement observed in this study in mice, the dose should be discussed. The dose in humans is 150 mg per m2 of body surface. Considering 0,5 m2 of surface per 10 kg of body weight (https://www.ouh.nhs.uk/oxparc/professionals/documents/Body-surfaceareaCCLGChart1.pdf) a dose of 75 mg represent around 7.5 mg/kg. This is very far from the 450 mg/kg indicated in this paper. This should be discussed, since a possible limitation of the study is whether a dose more close to the clinically used in patients (ie, 7.5 mg/kg) is able to produce cardiovascular improvements in mice.

Thank you for revealing that our description was not clear, probably because of a misplaced comma (now corrected). The dose was 450 mg of lonafarnib per kg of food (not per kg of mouse body mass), now stated explicitly. Normal mice can eat up to 10% of their body mass per day, though the continued loss of body mass in progeria mice suggests lower levels of daily consumption. At the maximum expected level (10% of body mass per day), each progeria mouse could ingest 0.675 mg of lonafarnib per day, which would translate into 45 mg/kg per mouse maximum. At half that level, this would translate into 22.5 mg/kg, much closer to that stated above for children (7.5 mg/kg). This is now stated clearly.

Note, in addition, that we received the lonafarnib as an in-kind gift from the Progeria Research Foundation (PRF) under formal agreement and we used their recommended dosage, per their request (now noted). PRF is the international leader in directing progeria research and their opinion was valued, their request was respected.

Finally, we also now note (in Discussion) that additional doses as well as periods of treatment and ages of initiation of treatment would be important to evaluate in the future. As an example, we now include a lonafarnib treatment arm during which drug was given from P100 to P168 (in contrast to the originally and still reported P21 to P168, which yielded better outcomes). For completeness, we now report and compare both of these treatment arms.

– Since no significant improvement in body mass is observed (line 117, page 4) it should be discussed whether lonafarnib administered to wild type mice affects to body weight.

It would seem that studies of the effect of lonafarnib on body mass in wild-type mice would have been warranted if treated progeria mice had either increased or decreased body mass. Since there was no change in body mass in treated progeria mice, we did not feel that it was important to test this lack of an effect in wild-type mice.

– The paper indicates that lonafarnib improves aortic composition, in particular lowering accumulation of mural proteoglycans (line 142, page 5, and Figures 4B and 4C). It would be nice to mark/highlight in the corresponding figures the area or the specific zone in the image where these improvements can be appreciated.

We agree and now present a much improved and expanded new figure while placing the prior figure in the Supplement since no two vessels were identical. We trust that these changes address this important issue, thank you.

Note, in addition, that all assessments are based on validated custom software-based quantification of many sections (the software is available via github). We tried to show representative images, but there is considerable spatial variability across specimens. For this reason, we previously presented a novel computational model to examine potential effects of stocastically distributed PGs/GAGs (Murtada et al., 2020). We have revised the data and presentation and have tried to improve the current presentation.

– In relationship with the former result, is there any insight on the mechanism by which lonafarnib reduces proteoglycan accumulation?

Despite an increasing data base on vascular changes in progeria, there is currently no well accepted mechanism by which proteoglycans accumulate in the large arteries and thus it would be difficult to determine mechanism of improvement. Our focus was, rather, on potential systemic effects on heart and large artery function, which have more clinical relevance.

– Considering the importance of the number of vascular smooth muscle cells (VSMCs) in progeria (Cells 2020, 9, 656; Circulation. 2018, 138, 266; eLife 2020, 9, e54383) this parameter should be also analyzed.

Agreed and corrected; we now show in the revised figure 4 that medial cell mass decreased in the aorta with progeria and improved slightly with lonafarnib, yet most importantly there was no improvement in aortic vasoconstriction (now shown explicitly in a new figure) though there was improvement in the muscular artery. It is, of course, function that is most important.

– In page 9 (lines 257-260) the authors put in context the obtained results regarding survival with other treatments. However, there is one study that should be also incorporated, since 174 days are obtained in 50% survival extension after administration of an ICMT inhibitor (ACS Central Sci. 2021, 7, 8, 1300). This work should be also cited.

Agreed, this omission has been corrected.

– References are difficult to find, they should be listed in alphabetical order.

Agreed and corrected.

Reviewer #2 (Recommendations for the authors):This paper focuses on the effects of the drug Lonafarnib, which is a farnesyltransferase inhibitor used as a therapeutic for patients with Hutchinson-Gilford progeria syndrome (HGPS) an accelerated aging disease characterized by premature death due to cardiovascular problems. Lonafarnib significantly increases lifespan, but it's effects on cardiac function have not been documented. The authors used an established mouse model for HGPS that carry the point mutation of the LMNA gene exhibiting severe cardiovascular disease and shortened life span. Homozygous G609G mice show progressive thoracic aorta structural stiffening and significant reduction in biomechanical function. Lonafarnib significantly improved survival, left ventricular diastolic function and descending thoracic aortic pulse-wave velocity, with lower accumulation of mural proteoglycans and calcification in the aorta. Studies also showed a moderate ability of Lonafarnib to prevent decline in vascoconstrictive and vasodilatory capacities of the muscular-type mesenteric artery. Finally, combined therapy with the MTOR inhibitor rapamycin was not beneficial, and even deleterious regarding the pulse wave velocity of the aorta. The authors conclude that Lonafarnib treatment can significantly improve both arterial structure and function to reduce pulse wave velocity and left ventricular diastolic function.StrengthsThis paper provides important information on Lonafarnib's effects as a therapeutic agent in HGPS. By maintaining the mouse model of HGPS on a specific gel diet, the authors were able to extend the lifespan of the mice, and thus the cardiovascular disease phenotype, without resorting to external parameters such as high-fat diet, which would have created a bias towards atherosclerosis and the metabolic syndrome. The paper is well written with the objectives stated clearly.

Thank you.

WeaknessesA question that arises is with regards to the combination therapy (Figure 2, Table S1). There is no data provided on the HGPS mice treated with rapamycin alone, thus making it difficult to understand how the authors can determine the reasons underlying the absence or worsening of effects of both Lonafarnib and rapamycin. If this data is available, it should be included. If not, it is questionable if this data as it brings additional value to the paper. It would also be helpful to the reader to present the reasoning behind dual treatment with Ioanafarnib and rapamycin in the introduction.

As the Reviewer likely knows, lonafarnib has shown benefits in progeria children but there are ongoing studies to try to identify combination therapies that improve outcomes further. One such study combines lonafarnib and an mTOR inhibitor. It was our goal, therefore, to test a combination therapy as well, noting that it will be unlikely that any future clinical trial will not include lonafarnib given the good results to date. Indeed, within the formal agreement we executed with the Progeria Research Foundtion (PRF) as part of their in-kind donation of lonafarnib they state: “…clinical trials testing new potential medications for HGPS will involve at least one combination treatment arm, administering lonafarnib plus a new drug of interest. Therefore PRF suggests that researchers consider similar preclinical studies, with one arm testing combination treatment with lonafarnib plus new compound of interest.” We respected this request and focused on lonafarnib + rapamycin. Because of the poor outcome, however, we obviously evaluated rapamycin alone – the results were similarly poor.

In an attempt to simplify the message of the initial paper, we had left out the set of data using rapamycin alone from P100 to P168. As requested, these data are now included as a survival curve. Given the poor outcomes, however, we did not perform complete cardiovascular assessments (indeed could not since most of the mice died prior to the study endpoint). Now noted.

There are some imprecise statements in the text in which the authors indicate changes in parameters, when in fact, the differences are not statistically significant. For example, lines 123-124 (Table 1) "Most dimensioned cardiac metrics, such as LV chamber volumes and stroke volume (SV), were lower in both untreated and lonafarnib-treated progeria mice…", when in fact it is not indicated as being statistically significant in Table 1. Also, in Figure 2F legend, it is indicated that lonafarnib improved cardioac output, when in the text, it is indicated as a trend toward improved cardiac output (line 129). The text should be reviewed carefully and corrected.

Thank you. We have tried to correct all inconsistencies and/or render the text precise. Regarding the latter, please note that we previously showed (Murtada et al., 2020) that calculated dimensioned cardiac metrics such as cardiac output and stroke volume and aortic metrics such as inner radius are statistically lower in progeria relative to wild-type controls, BUT these lower values are actually allometrically appropriate for the lower body mass (e.g., 15 g for progeria vs. 25-30 g for healthy wild-type), noting that allometric relations follow the nonlinear power-law form Y=bM^c^ where Y is any variable, M is body mass, and b and c are parameters. Hence, although it true to say “statistically lower” this would actually be misleading (since they are body mass appropriate, which is most often not appreciated), hence we have tried to avoid such situations and simply say lower (numerically). Note, too, that vascular metrics also depend on the values of pressure at which the metrics (e.g., stiffness) are evaluated, and they can differ when evaluated at group-specific versus common pressures. We report both in the Tables but prefer the common pressure comparison in the figures, which is now noted. Thank you for identifying these issues that would not be clear to most readers; they have now been addressed in the revision.

Figure 1. In Figure 1A, it would be helpful to include data on the WT littermates providing a reference to the change in structural stiffness as a function of age.

Agreed and added, though we only provide the WT data at P42 and P100 so as to not clutter the figure (after which the WT do not change for months – Murtada et al., 2021; Cavinato et al., 2021).

Figure 4. The authors report significantly decreased proteoglycan accumulation in the group that received lonafarnib. This is not evident based on the images. For the analysis, how did the authors normalize the values obtained, the amount of proteoglycan and calcification was relative to what parameter?

First, please note that our assessments are based on validated custom software-based quantification of many sections and we have found that the computer often detects changes that are difficult to visualize until guided by the section-dependent computation. We tried to show representative images, but there is considerable spatial variability across specimens. For this reason, we previously presented a novel computational model to examine potential biomechanical effects of stocastically distributed PGs/GAGs (Murtada et al., 2020). That said…

Thank you for this important comment, we agree. We have since re-confirmed our method of quantification and had a second investigator not involved in the original study re-analyze the entire data set de novo, resulting in a new Figure with updated values, which are not different from those reported previously hence the prior conclusions remain. We feel, however, that the visual appreciation of the effects of progeria and its treatment is much improved in this new figure.

Reviewer #3 (Recommendations for the authors):The authors have investigated the effect of lonafarnib, a farnesyltransferase inhibitor, in a mouse model of progeria. The treatment with lonafarnib extended lifespan in progeria mice, reduced arterial thickening, and improved arterial function in progeria mice. This treatment increases the lifespan of the mice and prevents, at least in part, the thickening, and the loss of contractility of the arteries using appropriate in vivo and in vitro tools. This study brings new information on the pathophysiology of progeria and on the mechanism of lonafarnib which improves arterial wall structure and contractility.1 – The data shown in figure 1 need to be reinforced. Indeed, in GG mice the aorta becomes translucent whereas its thickness increases to more than 100µm. Such a tremendous change in structure requires a more in-depth analysis. It is important to show not only the degree of calcification (this is shown / alizarin red) but also the location of the cells in the wall (and the type of cells). Is there any smooth muscle left? Which phenotype do they have, and which other cells have eventually infiltrated the vascular wall if any (or the adventitia which is very thick in GG mice). Figure 4 shows a more limited thickening of the aortic wall. Indeed, Movat staining shows a relatively well-structured media and a thicker adventitia. This phenotype suggests a strong inflammatory response of the vascular wall which could explain the loss of contractility in GG mice. This needs to be analyzed more in depth.

We have tried to reinforce the data in multiple ways. First, we have added to Figure 1A wild-type results at P42 to P100 for a direct comparison (which was previously found only in tabulated form in the supplemental table). Second, we completely reanalyzed all of our histological data (with an independent investigator not involved in the original study) and now provide a new Figure 4 that also shows whole histological cross-sections (not just portions) and shows Movat results for medial cell cytoplasm (new), medial collagen (new), medial elastin (improved), and medial PG/GAGs (improved). These representative images better reflect the results of the quantitation, also rechecked independently and found to be the same as originally reported.

Importantly, given that the extreme loss of vascular SMCs has been one of the most consistent observations in progeria studies of both patients and murine models, we now include in our new Figure data (on medial cell density) that are consistent with this observation. We also show in a revised Figure that there was no recovery of SMC function in the aorta with lonafarnib treatment (simply stated but not shown, before) despite less SMC loss whereas there was significant recovery in SMC function in small arteries. Thank you for this comment for these are important clarifications.

Yes, it would be interesting to assess all cell types, but it appears (and largely confirmed at the 11^th^ International Progeria Workshop at Boston in Nov 2022) that SMCs are most important – it was their function that we focused on and thus evaluated, noting that we also assessed endothelial cell function via Ach.

2 – Contractility and dilation: The strong reduction in contractility in the aorta and in mesenteric arteries suggests a major SMC defect. In parallel, endothelium-mediated (Ach) dilation is also abolished in mesenteric arteries suggesting that endothelial cells (ECs) are also affected. Ach-dependent dilation should also be measured in the aorta.

Now added in a revised figure. These are good comments, though note that a vessel cannot dilate ex vivo if it has not been pre-constricted, so loss of SMC contractility is definitive whereas loss of Ach-induced dilatation is not.

This loss of SMCs and ECs function suggests a more global effect of the disease on vascular cells. Interestingly, the treatment with lonafarnib prevents this loss of cellular function (and loss of cells?) in the mesenteric arteries. This is not shown in the aorta whereas this information seems essential. Indeed, it's important to have the reactivity data in figure 3 (as in figure 1 G, H and I + Ach).

Related to the above, now shown explicitly in addition to the prior brief comment in the text.

3 – Arterial diameter: In both the aorta and the mesenteric artery, arterial diameter is severely reduced in GG mice. As small arteries control tissues perfusion and pressure it is important to provide the blood pressure values in the different groups of mice. In the mesenteric arteries the inward remodeling resembles that seen in hypertension as discussed by the authors. The authors should measure tissue perfusion (laser Doppler of the mouse feet for example).

These are insightful comments. In this regard, however, please note that we previously showed (Murtada et al., 2020) that many cardiovascular quantities (e.g., aortic radius, LV mass, LV diameter, stroke volume, cardiac output) are “statistically lower” in progeria at P140 when assessed using standard statistics, BUT these lower values are actually allometrically appropriate for the lower body mass. Thus statistics based on age- (but not body mass-) matched mice can thus be misleading. There is, therefore, a need for caution when interpreting such data in growth compromised mice, which we try to do.

Nonetheless, as suggested by the Reviewer, it would be interesting to measure perfusion in different end-organs, but our focus was on cardiac function (the primary clinical diagnosis) and related changes in the aorta (with a comparison to a representative muscular artery) given all that we know about central arterial stiffening and cardiac dysfunction in natural aging.

We routinely measure central blood pressures using Millar catheters following our echo measurements, but these measurements were not successful in the progeria mice (so small and fragile, especially under anesthesia), hence these data were either unavailable or unreliable and are not reported.

In addition, an histological analysis of the myocardium is necessary to assess whether hypertrophy or excessive dilation occurs in GG mice with arterial thickening.

We (Murtada et al., 2020), and others, previously reported histological assessments of the myocardium (we at P140) and did not see any overt pathology consistent with the preserved ejection fraction (EF) and fractional shortening (FS). Given that we did not see any decrements in EF or FS at P168, we focused on measures of possible diastolic dysfunction and associated changes in central artery function.

4 – Lonafarnib did not prevent this inward remodeling whereas it improved contractility and dilation in MA (how about the aorta?). Again, blood pressure and tissue perfusion are necessary to understand the effect of the treatment.

Again, smaller diameters need not imply inward remodeling; we showed previously that multiple metrics scale allometrically (including smaller diameters for smaller mice as they should). As noted above, we were not successful in obtaining consistent/reliable blood pressure measurements in the progeria mice and thus focused on non-invasive echocardiographic metrics of function. We have added the lack of recovery of vasoactive function of the aorta (previously simply stated) in a revised figure.